# Siglec-9 defines and restrains a natural killer subpopulation highly cytotoxic to HIV-infected cells

Opeyemi S. Adeniji[1], Leticia Kuri-Cervantes[2], Chenfei Yu[3], Ziyang Xu[1,2], Michelle Ho[1], Glen M. Chew[4], Cecilia Shikuma[4], Costin Tomescu[1], Ashley F. George[5,6], Nadia R. Roan[5,6], Lishomwa C. Ndhlovu[4,7], Qin Liu[1], Kar Muthumani[1], David B. Weiner[1], Michael R. Betts[2], Han Xiao[3], Mohamed Abdel-Mohsen[1] *

1 The Wistar Institute, Philadelphia, Pennsylvania, United States of America, 2 University of Pennsylvania, Philadelphia, Pennsylvania, United States of America, 3 Rice University, Houston, Texas, United States of America, 4 University of Hawaii, Honolulu, Hawaii, United States of America, 5 Gladstone Institutes, San Francisco, California, United States of America, 6 University of California San Francisco, San Francisco, California, United States of America, 7 Weill Cornell Medicine, New York, New York, United States of America

* mmohsen@Wistar.org

**Data Availability Statement:** All relevant data are within the manuscript and its Supporting Information files.

## Abstract

Siglec-9 is an MHC-independent inhibitory receptor expressed on a subset of natural killer (NK) cells. Siglec-9 restrains NK cytotoxicity by binding to sialoglycans (sialic acid-containing glycans) on target cells. Despite the importance of Siglec-9 interactions in tumor immune evasion, their role as an immune evasion mechanism during HIV infection has not been investigated. Using *in vivo* phenotypic analyses, we found that Siglec-9+ CD56dim NK cells, during HIV infection, exhibit an activated phenotype with higher expression of activating receptors and markers (NKp30, CD38, CD16, DNAM-1, perforin) and lower expression of the inhibitory receptor NKG2A, compared to Siglec-9- CD56dim NK cells. We also found that levels of Siglec-9+ CD56dim NK cells inversely correlate with viral load during viremic infection and CD4+ T cell-associated HIV DNA during suppressed infection. Using *in vitro* cytotoxicity assays, we confirmed that Siglec-9+ NK cells exhibit higher cytotoxicity towards HIV-infected cells compared to Siglec-9- NK cells. These data are consistent with the notion that Siglec-9+ NK cells are highly cytotoxic against HIV-infected cells. However, blocking Siglec-9 enhanced NK cells' ability to lyse HIV-infected cells, consistent with the known inhibitory function of the Siglec-9 molecule. Together, these data support a model in which the Siglec-9+ CD56dim NK subpopulation is highly cytotoxic against HIV-infected cells even whilst being restrained by the inhibitory effects of Siglec-9. To harness the cytotoxic capacity of the Siglec-9+ NK subpopulation, which is dampened by Siglec-9, we developed a proof-of-concept approach to selectively disrupt Siglec/sialoglycan interactions between NK and HIV-infected cells. We achieved this goal by conjugating Sialidase to several HIV broadly neutralizing antibodies. These conjugates selectively desialylated HIV-infected cells and enhanced NK cells' capacity to kill them. In summary, we identified a novel, glycan-based

**Funding:** This work is supported by the NIH grants R01 AI165079 to M.A-M and H.X and R21AI143385 to M.A-M. Additional support was also provided by NIH grants P30 AI 045008 and UM1AI164570 as well as the Robert I. Jacobs Fund of The Philadelphia Foundation. H.X. is supported by NIH grants R35 GM133706 and R21 CA255894, U.S. Department of Defense grant W81XWH-21-1-0789, Cancer Prevention & Research Institute of Texas (CPRIT) grant RR170014, and the Robert A. Welch Foundation grant C-1970. The funders had no role in study design, data collection, and analysis, decision to publish, or preparation of the manuscript.

**Competing interests:** The authors have declared that no competing interests exist.

interaction that may contribute to HIV-infected cells' ability to evade NK immunosurveillance and developed an approach to break this interaction.

## Author summary

The Siglec-9 molecule, expressed on NK cells, binds to Sialic acid, expressed on target cells, and this binding induces an inhibitory signal to NK cells. As such, Siglec-9 functions as a "glyco-immune negative checkpoint". Despite the importance of such Siglec-9-Sialo-glycan interactions in tumor immune evasion, their role as an immune evasion mechanism during HIV infection has not been investigated. We found that the cytotoxicity of the Siglec-9+ CD56dim NK subpopulation against HIV-infected cells is indeed being restrained by the inhibitory nature of the Siglec-9 molecule itself. However, we also found that this Siglec-9+ CD56dim NK subpopulation is highly cytotoxic against HIV-infected cells compared to the Siglec-9- CD56dim NK subpopulation. Our data suggest that Siglec-9 is expressed on highly cytotoxic NK cells, where it restrains their high cytotoxicity. We have also developed a proof-of-concept immunotherapy approach to selectively disrupt Siglec/sialoglycan interactions between NK cells and HIV-infected cells. We did so by conjugating Sialidase to HIV broadly neutralizing antibodies. These conjugates selectively desialylated HIV-infected cells and enhanced NK capacity to kill infected cells. Our findings bring to light the potentially relevant and previously unrecognized glyco-immune checkpoint mechanisms that may contribute to the ability of HIV-infected cells to evade host immunosurveillance.

## Introduction

The barrier to HIV eradication is the ability of the virus to establish persistent infection in CD4+ T cells and possibly other cell types [1–7]. A "functional HIV cure" may be established by enabling antiretroviral therapy (ART)-independent suppression of HIV [8]. One proposed approach to reach this goal is "shock and kill" [9]. In this approach, latency reversal agents (LRAs) are administered to reverse HIV latency and induce viral production; however, reversing latency is only the first step (shock). The second step (kill) requires efficient immune responses to clear reactivated cells. Clinical trials involving LRAs have shown that immune responses of HIV-infected ART-treated individuals cannot clear reactivated reservoirs, suggesting that adjuvant immunotherapy is needed [10–16]. One potential adjuvant strategy is to enhance the cytotoxicity of natural killer (NK) cells during viral reactivation (achieved by LRAs or by ART-cessation). Developing a strategy to achieve this goal would require a better understanding of the factors that determine NK functions against HIV-infected cells.

The functions of NK cells can be influenced by the cell-surface glycosylation of their target cells. NK cells express several cell-surface lectins (glycan-binding proteins), including two belonging to the Siglec family: Siglec-7 and Siglec-9. Siglecs (Sialic acid-binding immunoglobulin-type lectins) are immunoreceptor tyrosine-based inhibitory motif (ITIM)-containing, MHC-independent inhibitory receptors that control host immune responses by interacting with Sialic-acid containing glycans on the surface of target cells. Siglec-7 is expressed on almost all NK cells and binds to α2–8 Sialic acid. Decreased levels of Siglec-7 have been described as a marker for dysfunctional NK cells in HIV viremic individuals [17–19]. By contrast, Siglec-9 is selectively expressed on a subset of the CD56dim NK cells (the cytolytic subset of NK cells) [20] and binds to α2–3 Sialic acid. The binding of Siglec-9 to α2–3 Sialic acid on target cells induces

an inhibitory signal transduction cascade by recruiting the tyrosine phosphatase SHP-1, which counteracts the phosphorylation-mediated activation of other signaling molecules [21,22]. As such, Siglec-9 functions as a glyco-immune negative checkpoint, analogous to the PD1 checkpoint on activated CD8+ T cells.

Siglec-9 continues to transmit inhibitory signals into NK cells even when target cells have lost the expression of MHC class I molecules (missing-self) or when the classical inhibitory NK receptors are inefficiently engaged [23]. This MHC independence of the Siglec-9 molecule makes it a good target for exploitation by cancer cells or virally-infected cells to evade host immune surveillance. Indeed, emerging evidence suggests that the Siglec-9+ CD56dim NK population plays an important role in regulating NK cytotoxicity against cancer cells and hepatitis B virus (HBV)-infected cells [24,25]. However, the role of Siglec-9 expression on NK cells during HIV infection has not been examined.

In this study, we investigated the role of Siglec-9+ CD56dim NK cells in controlling HIV infection using both *in vivo* phenotypic analyses of samples from HIV-infected individuals and controls, as well as *in vitro* cytotoxicity assays. Our data show that the Siglec-9+ CD56dim NK cell subpopulation has high cytolytic activity against HIV-infected cells, likely due to its elevated expression of several NK activating receptors and low expression of inhibitory receptors. However, the inhibitory nature of the Siglec-9 molecule restrains the cytolytic activity of these cells, which would otherwise be even more cytotoxic. We also developed and validated a proof-of-concept approach to selectively disrupt the Siglec/sialoglycan inhibitory interactions between NK and HIV-infected cells.

## Results

### Siglec-9 is expressed on a subset of activated CD56dim NK cells during HIV infection

A decreased level of Siglec-7 has been described as a marker for a dysfunctional NK subset in HIV viremic individuals [17–19]. However, the role of Siglec-9 in HIV infection has not been elucidated. We first sought to characterize the cell-surface expression of Siglec-9 on NK cells and determine whether Siglec-9 expression levels differed between HIV+ (ART-suppressed or viremic) individuals and HIV-negative controls (clinical data of this cohort are in **S1 Table**). We performed a comprehensive 27-color phenotypic analysis (gating strategy is in **S1 Fig**) of Siglec-9+ CD56dim NK cells. First, we found that Siglec-9 is expressed on a subset of CD56dim NK cells regardless of HIV status (**Fig 1A**). Next, we found that the levels of Siglec-9+ CD56dim NK cells are significantly lower in HIV+ individuals (viremic or ART-suppressed) compared to HIV-negative controls (**Fig 1B–1D**). As several studies have reported the upregulation of HLA-DR on NK cells during HIV infection [26,27], we examined the potential impact of including HLA-DR+ cells in our analysis. Our findings were not impacted by including or excluding HLA-DR+ cells (**S2 Fig**). Given the focus of our analyses on the cytolytic CD56dim NK cells and the high expression of Siglecs on monocytes [20,22], we elected to exclude HLA-DR+ cells from subsequent analyses, to ensure the exclusion of monocytes. To further ensure the exclusion of monocytes from our analysis, we examined whether Siglec-9+ CD56dim NK cells express CD7, as monocytes do not express CD7 [28]. Indeed, the vast majority of Siglec-9+ CD56dim NK cells express CD7 as shown in **S3 Fig**. These data suggest that HIV infection, regardless of the treatment status, is associated with a depletion of the Siglec-9+ CD56dim NK cells.

The cytotoxic potential of NK cells is regulated by a collection of activating and inhibitory signals delivered by cell surface receptors [29]. We, therefore, evaluated Siglec-9+ CD56dim NK cells for their expression of activation and inhibitory receptors/markers during HIV infection. We evaluated the expression of 18 markers on Siglec-9+ CD56dim NK cells (**Fig 2A**). We then

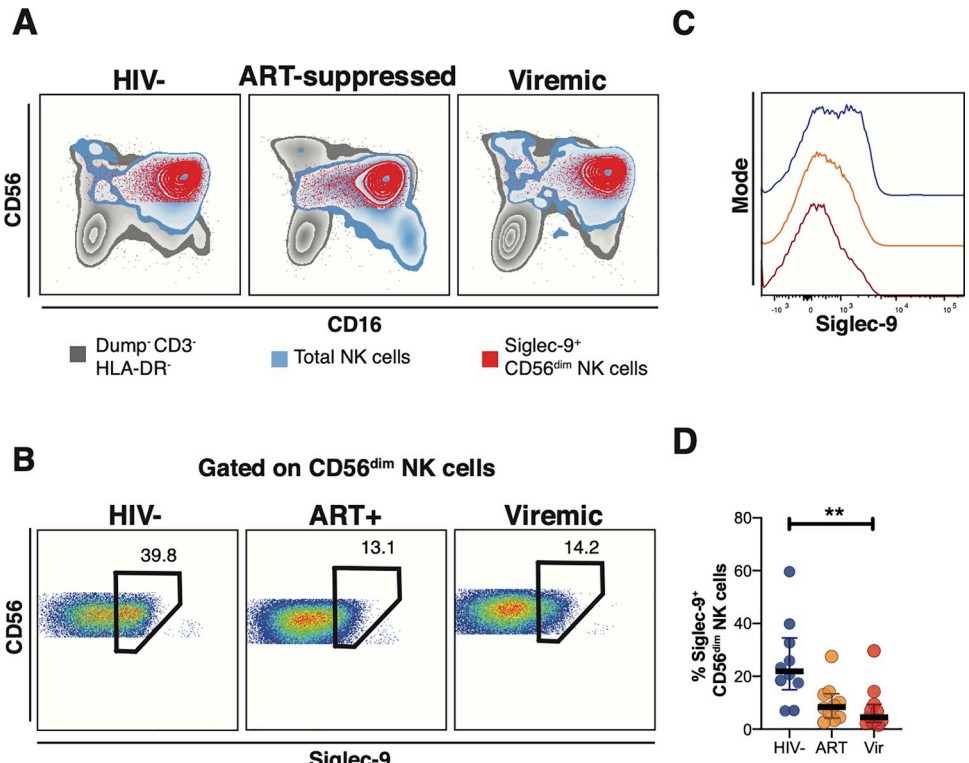

**Fig 1. Expression of Siglec-9+ CD56dim NK cells during HIV infection. (A)** Overlay plots showing the distribution of Siglec-9+ CD56dim NK cells (red) compared with total NK cells (blue) and non-T cell lymphocytes (grey). **(B-C)** Representative plots showing the frequency **(B)** and expression (MFI overlay) **(C)** of Siglec-9 in total CD56dim NK cells in HIV- (blue line), HIV+ ART+ (orange line), and HIV+ viremic (red line) individuals. **(D)** Decreased frequency of Siglec-9+ CD56dim NK cells during HIV infection compared to HIV- controls. Lines in graphs indicate the median of the group. ** $p < 0.01$. Mann-Whitney rank test was used to compare between groups. n = 10 HIV-negative controls, 11 HIV+ viremic, and 10 HIV+ on suppressive ART.

compared the expression of these markers on Siglec-9+ versus Siglec-9- CD56dim NK cells (**Fig 2B and 2C**). We found that Siglec-9+ CD56dim NK cells, during HIV infection, exhibit higher expression of several NK activating/cytotoxic receptors and markers including CD16 (% and mean fluorescence intensity (MFI)), CD38 (% and MFI), NKp30 (% and MFI), DNAM-1 (% and MFI), and perforin (%); and lower expression of the inhibitory receptor NKG2A (MFI) and TIGIT (% and MFI), compared to Siglec-9- CD56dim NK cells (**Fig 2B and 2C**). However, Siglec-9+ CD56dim NK cells also express higher levels of the inhibitory markers Siglec-7 (% and MFI) and KIR3DL1 (% and MFI). Together, these data suggest that Siglec-9 marks a distinct subpopulation of NK cells during HIV infection, characterized by high expression of several NK activating receptors and markers and differential expression of several inhibitory receptors and markers. These data are in agreement with existing literature from cancer and HBV fields that the Siglec-9+ CD56dim NK subpopulation harbors a mature and activated phenotype [24,25].

## The frequency of Siglec-9+ CD56dim cells correlates with lower viral load during viremic HIV infection and lower HIV DNA during ART-suppressed HIV infection

Given the potentially activated phenotype of Siglec-9+ CD56dim NK cells during HIV infection, we next examined the relationship between Siglec-9 expression on NK cells and plasma viral

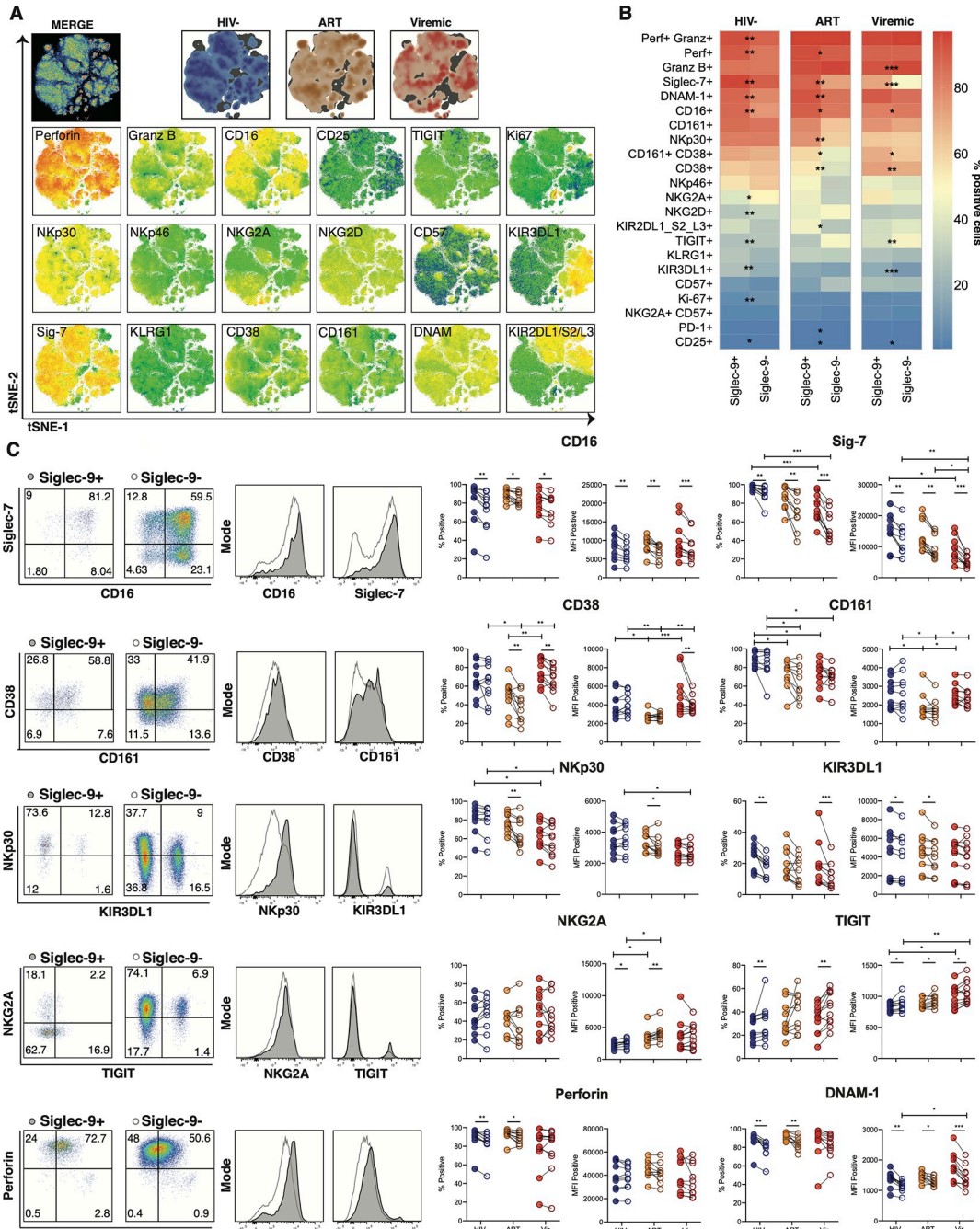

**Fig 2. The phenotype of Siglec-9⁺ CD56^dim NK cells. (A)** Global t-SNE visualization of Siglec-9⁺ CD56^dim NK cells for all individuals pooled, with Siglec-9⁺ CD56^dim NK cells from HIV-, HIV+ ART+, and HIV+ viremic individuals concatenated and overlayed (dimensionality reduction performed from 234,000 cells in 21 dimensions, 10,000 iterations, excluding parameters used to define the population: time, FSC, SSC, viability, CD14, CD19, CD3, and Siglec-9). Bottom: t-SNE projections of the 18 indicated proteins expression. **(B)** Heatmaps showing the percentages of Siglec-9⁺ and Siglec-9⁻ CD56^dim NK cells expressing the indicated activation and inhibitory markers in HIV-, HIV+ ART+, and HIV+ viremic individuals. **(C)** Comparative analyses of frequency (% positive) and expression (MFI of the positive population) of CD16, Siglec-7, CD38 CD161, NKp30, KIR3DL1, NKG2A, TIGIT, Perforin, and DNAM-1 on Siglec-9⁺ vs. Siglec-9⁻ CD56^dim NK cells. Left: Representative flow plots and histograms from HIV+ ART-suppressed donors are shown. Numbers inside the plots represent the gated percentage within the parent population. Mann-Whitney rank test was used to compare between groups. Paired Wilcoxson test was used to compare Siglec-9⁺ and Siglec-9⁻ within each group. ***p<0.001, ** p<0.01, *p<0.05. n = 10 HIV-negative controls, 11 HIV+ viremic, and 10 HIV+ on suppressive ART.

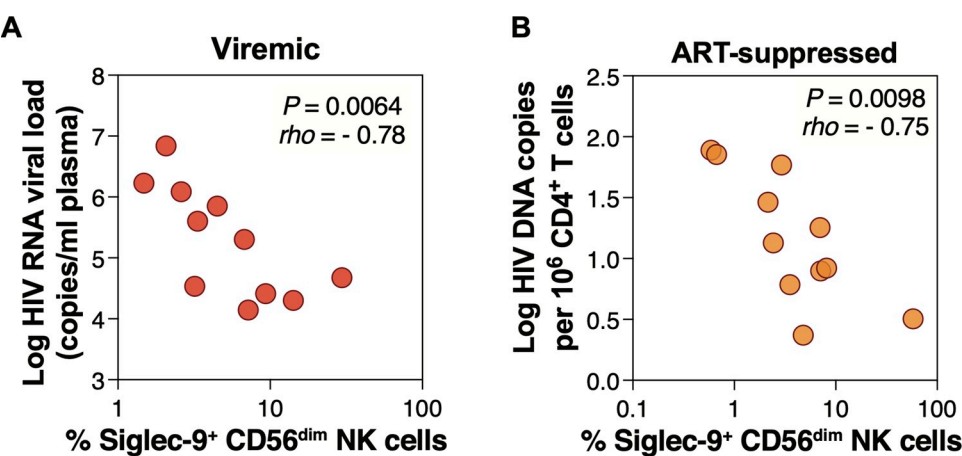

**Fig 3. Frequency of Siglec-9<sup>+</sup> CD56<sup>dim</sup> NK cells correlates with viral load during viremic HIV infection and levels of CD4<sup>+</sup> T cell-associated HIV DNA during ART-suppressed HIV infection. (A)** Spearman correlation between the frequency of Siglec-9<sup>+</sup> CD56<sup>dim</sup> NK cells and HIV plasma viral load during viremic HIV infection. n = 11. **(D)** Spearman correlation between the frequency of Siglec-9<sup>+</sup> CD56<sup>dim</sup> NK cells and cell-associated HIV DNA copies per million CD4<sup>+</sup> T cells during ART-suppressed HIV infection. n = 11.

load during HIV viremic infection. We found that the percentage of Siglec-9<sup>+</sup> CD56<sup>dim</sup> NK cells correlated with lower HIV plasma viral load (**Fig 3A**). We next examined the relationship between Siglec-9 expression on NK cells and total HIV DNA measured in CD4<sup>+</sup> T cells from 11 HIV+ ART+ individuals (clinical data of this cohort are in **S2 Table,** and gating strategy is in **S4 Fig**). We found that the percentage of Siglec-9<sup>+</sup> CD56<sup>dim</sup> NK cells correlated with lower levels of CD4<sup>+</sup> T cell-associated total HIV DNA (**Fig 3B**). In contrast, we did not observe significant correlations between the percentage of Siglec-7 on CD56<sup>dim</sup> or CD56<sup>bright</sup> NK cells and total HIV DNA. These data, together with data in Figs 1 and 2, are consistent with the notion that Siglec-9<sup>+</sup> CD56<sup>dim</sup> NK cells are cytotoxic and may play a role in controlling HIV infection, in line with their ability to control infection by other viruses [25].

## Siglec-9<sup>depleted</sup> NK cells exhibit lower cytotoxicity towards HIV+ cells compared to total NK cells

The phenotypic data in Figs 1–3 suggest that Siglec-9<sup>+</sup> NK cells may be highly cytotoxic. To test this, we compared the cytotoxicity of NK cells depleted of the Siglec-9<sup>+</sup> population (Siglec-9<sup>depleted</sup> NK cells) to the cytotoxicity of total NK cells against HIV-infected targets. We isolated NK cells from several HIV-uninfected donors and depleted Siglec-9<sup>+</sup> NK cells (**Fig 4A**). We then compared the cytotoxicity of total and Siglec-9<sup>depleted</sup> NK cells against a T cell line, HUT78, infected with SF2 HIV (dual tropic virus) (HUT78/SF2; **S5A Fig**). These T cells have levels of Siglec-9 ligands (α2–3 Sialic acid) comparable to primary human CD4<sup>+</sup> T cells (**S5B Fig**). We assessed cytotoxicity using three different measures: (1) NK degranulation [frequency of CD56<sup>dim</sup> NK cells expressing CD107a and IFN-γ; by flow cytometry [30–32]]; (2) Levels of lactate dehydrogenase (LDH) released into the supernatant from damaged cells [normalized to background using target cells and effector cells alone; by luminescence assay]; and (3) the proportion of lysed target cells using the CFSE/SYTOX method [33]. In the CFSE/SYTOX method, target cells were pre-labeled with CFSE dye. After co-culturing effector and target cells, killed target cells were identified by SYTOX Red stain, which selectively permeates dead cells. Cytotoxicity was measured as the proportion of dead target cells (SYTOX Red<sup>+</sup> CFSE<sup>+</sup>) to the total number of targets (CFSE<sup>+</sup>); by flow; normalized to target cells only [34].

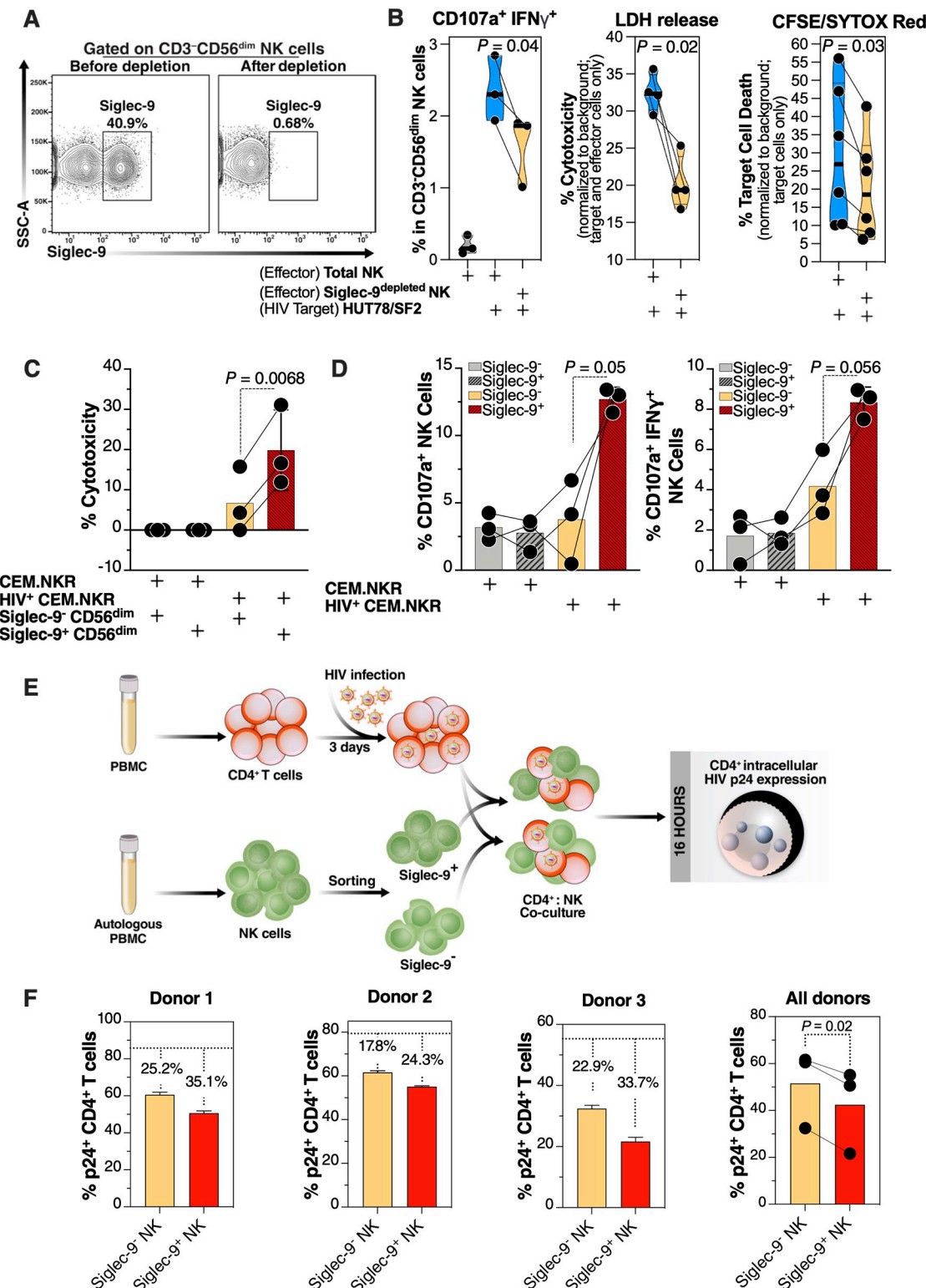

**Fig 4. Siglec-9+ CD56dim NK cells exhibit higher cytotoxicity towards HIV+ cells compared to Siglec-9- CD56dim NK cells. (A)** A representative example of depletion of Siglec-9+ NK cells. **(B)** Siglec-9depleted NK cells exhibit lower cytotoxicity towards HIV-infected HUT78/SF2 targets compared to total NK cells. Cytotoxicity was assessed using NK degranulation, left panel (n = 3 donors; E:T = 4:1), LDH release, middle panel (n = 4 donors; E:T = 10:1), and CFSE/SYTOX Red assay, right panel (n = 6 donors; E:T = 10:1). NK degranulation measured as CD107a+ IFNγ+. Assays from each donor were performed in 2–4 replicates, and the

average of these replicates per donor was used for statistical analyses. Statistical analyses were performed using paired t-tests. **(C-D)** FACS sorted Siglec-9$^+$ CD56$^{dim}$ NK cells exhibit higher cytotoxicity towards HIV+ CEM.NKR targets compared to Siglec-9$^-$ CD56$^{dim}$ NK cells. **(C)** Cytotoxicity was assessed using LDH release assay (n = 3 donors, E:T = 10:1). **(D)** Analysis of NK degranulation (n = 3 donors; E:T = 4:1) was made on total NK cells gated on Siglec-9$^+$ or Siglec-9$^-$ CD56$^{dim}$ NK cell subsets. Siglec-9$^+$ = Siglec-9$^+$ CD56dim NK cells and Siglec-9$^-$ = Siglec-9$^-$ CD56$^{dim}$ NK cells. Statistical analyses were performed using paired t-tests. **(E)** A schematic representation of the workflow to evaluate the cytotoxic potential of Siglec-9$^+$ and Siglec-9$^-$ CD56$^{dim}$ NK cells against autologous HIV-infected CD4$^+$ T cells. CD4$^+$ T cells were isolated from fresh PBMC and exposed to HIV-1 IIIB for 72 h. On the third day, effector NK cells were isolated from PBMC of the same donor, FACS sorted, and co-cultured with autologous HIV-infected CD4$^+$ T cells for 16 h. Following overnight incubation, the mixtures were stained for live/dead viability, CD3, and intracellular p24. **(F)** Data from the experimental design shown in (D). Dashed lines denote the percentage of p24$^+$ cells in control HIV-infected CD4$^+$ T cells cultured without effector cells. Percentages are percent reduction from dashed line. Assay from each donor was performed in triplicate (E:T = 10:1; n = 3 donors). Statistical analysis was performed using paired t-test.

Assays were performed in triplicate for each donor, and an average was used for statistical analysis. Results from every measure demonstrated that total NK cells exhibit higher cytotoxicity than Siglec-9$^{depleted}$ NK cells (**Fig 4B**), suggesting that the Siglec-9$^+$ CD56$^{dim}$ subpopulation of NK is an important contributor to NK cytotoxicity against HIV-infected cells.

## Siglec-9$^+$ NK cells exhibited higher cytotoxicity towards HIV+ cells compared to Siglec-9$^-$ NK cells

We next sorted Siglec-9$^+$ and Siglec-9$^-$ CD56$^{dim}$ NK (using fluorescence-activated cell sorting (FACS)) from several healthy donors (assays were done in triplicate for each donor, and an average was used for statistical analysis) and compared their cytotoxicity against the CEM. NKR cells (which are naturally resistant to NK killing without HIV infection) after infecting the CEM.NKR cells with DH12 HIV (a dual-tropic virus) (**S5C Fig**). Following incubation with HIV-infected cells, Siglec-9$^+$ CD56$^{dim}$ NK cells exhibited higher cytotoxicity (**Fig 4C**) and were more degranulated (**Fig 4D**) compared to Siglec-9$^-$ CD56$^{dim}$ NK cells. These data are in contrast with a recent publication by Jandus et al. [24], where Siglec-9$^+$ CD56$^{dim}$ NK cells exhibited lower activation against the K562 cancer cell line compared to Siglec-9$^-$ CD56$^{dim}$ NK cells. To examine if the cytotoxicity of the Siglec-9$^+$ CD56$^{dim}$ NK cells is target-dependent, we tested the capacity of Siglec-9$^+$ or Siglec-9$^-$ CD56$^{dim}$ NK cells from several healthy donors against the K562 cancer cell line as target cells (**S6 Fig**). Indeed, results from these assays using the K562 cancer cell line were in agreement with data from Jandus et al. [24], suggesting that the cytotoxicity of the Siglec-9$^+$ CD56$^{dim}$ NK subpopulation is target-specific.

We next examined the cytotoxic capacities of Siglec-9$^+$ and Siglec-9$^-$ CD56$^{dim}$ NK cells against autologous HIV-infected CD4$^+$ T cells (infected with HIV-1 IIIB) (**Fig 4E**). In these experiments, we used HIV-infected primary CD4$^+$ T cells as target cells and autologous NK cells as effector cells (**Fig 4E**). We examined the ability of these effector cells to reduce the levels of HIV-infected cells in these co-cultures as measured by intracellular p24 levels. This experiment was independently repeated three times, using cells from three donors, and each repeat was performed in multiple replicates. Following incubation with autologous HIV-infected CD4$^+$ T cells, Siglec-9$^+$ CD56$^{dim}$ NK cells exhibited a greater ability to reduce the levels of HIV-infected cells compared to Siglec-9$^-$ CD56$^{dim}$ NK cells (**Fig 4F**). Together, data from Fig 4 suggest that Siglec-9$^+$ CD56$^{dim}$ NK cells are highly cytotoxic against HIV-infected cells.

## Blocking Siglec-9 enhances the ability of NK cells to kill HIV-infected cells

The preceding data suggest that the Siglec-9$^+$ subset of NK cells may play a role in controlling HIV infection; however, the Siglec-9 molecule itself is an inhibitory receptor which can restrain its cytolytic capacity. Intriguingly, we found that blocking Siglec-9 (using an in-house Siglec-9 blocking antibody) enhanced the ability of donors' NK cells to kill HIV-infected cells

(HUT78/SF2 cells) (**Fig 5A**). These effects were not observed using Siglec-9[depleted] NK cells, demonstrating that the antibody enhances NK cytotoxicity against HIV+ cells by specifically blocking Siglec-9. Similar results were obtained using CEM.NKR cells infected with DH12 HIV (**Fig 5B and 5C**) in that sorted Siglec-9[+] CD56[dim] NK cells showed higher cytotoxicity (**Fig 5B**) and degranulation (**Fig 5C**) in the presence of the Siglec-9 blocking antibody.

We next examined the ability of the Siglec-9 blocking antibody to enhance NK cytotoxicity against autologous HIV-infected CD4[+] T cells (infected with HIV-1 IIIB) (**Fig 5D**). In these experiments, we used isolated HIV-infected primary CD4[+] T cells as target cells and autologous NK cells as effector cells (**Fig 5D**). This experiment was independently repeated three times, using cells from three donors, and each repeat was performed in multiple replicates. The Siglec-9 blocking antibody enhanced NK degranulation, in a manner selective to Siglec-9[+] CD56[dim] NK cells (**Fig 5E**). The Siglec-9 blocking antibody also reduced the levels of HIV-infected cells, as measured by intracellular p24 staining, compared to an isotype control antibody (**Fig 5F**). In addition to examining levels of HIV infection by intracellular p24 staining (**Fig 5F**), we also examined the titer of HIV in the co-culture supernatants using TZM-bl cells (cells with an HIV Tat-responsive long terminal repeat (LTR) promoter driving the expression of beta-galactosidase and firefly luciferase). Supernatants from co-cultures that contained the Siglec-9 blocking antibody had lower levels of infectious HIV, as evident by their lower ability to infect TZM-bl cells, compared to co-cultures that contained the isotype control (**S7 Fig**). Together, these data support a model in which Siglec-9[+] CD56[dim] NK cells help to control HIV infection but at the same time are being restrained by the inhibitory nature of Siglec-9 receptor signaling.

## Generation of HIV antibody-sialidase conjugates selectively disrupt Siglec/sialoglycan interactions between NK cells and HIV-infected cells

Many cells, not just HIV-infected ones, express sialoglycans, and Siglec-sialic acid interactions are important immune negative checkpoints against autoimmunity [35–38]. Targeted approaches have recently been developed in the cancer field [39,40] whereby conjugating sialidase (the enzyme that removes Sialic acid from glycans) to trastuzumab (Herceptin; an antibody against HER2[+] breast cancer cells) selectively desialylated HER2[+] breast cancer cells. This trastuzumab-sialidase conjugate prevented Siglec/Sialic acid-binding (both Siglec-7 and Siglec-9) and enhanced anti-tumor NK activity against HER2[+] but not HER2[−] cells [39]. Importantly, in an *in vivo* mouse model of breast cancer, antibody-sialidase conjugates were safe, effective and exhibited the low off-target activity and the high chemical stability needed for *in vivo* use [40].

To develop an approach that selectively targets HIV-infected cells, we employed a similar strategy by conjugating Sialidase to HIV broadly neutralizing antibodies (bNAbs). First, we used DNA constructs encoding three HIV bNAbs (3BNC117, PGT151, and NIH45-46) to produce antibodies using Expi293F cells [41]. Purified antibodies bind efficiently to HIV+ cells *in vitro* (**S8 Fig**). We then employed the proximity-induced antibody labeling (pClick) technology [42–44] to conjugate these antibodies to Sialidase (from *Salmonella typhimurium;* STSia). pClick allows site-specific labeling of native antibodies with payloads under mild conditions, thus minimizing the disruption of antigen and Fc receptor binding. This approach was recently proven safe and effective *in vivo* [44]. To site-specifically conjugate the HIV bNAbs with Sialidase using pClick, we first genetically incorporated 4-fluorophenyl carbamate lysine into the Glu25 position of a FB fused with Sialidase using the genetic code expansion technology (**Fig 6A top**). Next, we prepared bNAb-STSia conjugates by incubating bNAbs with 16 equivalents of FB-STSia for 48 h. We characterized the conjugates by SDS-PAGE and determined an enzyme/antibody ratio of 1.0 (**Fig 6A bottom**).

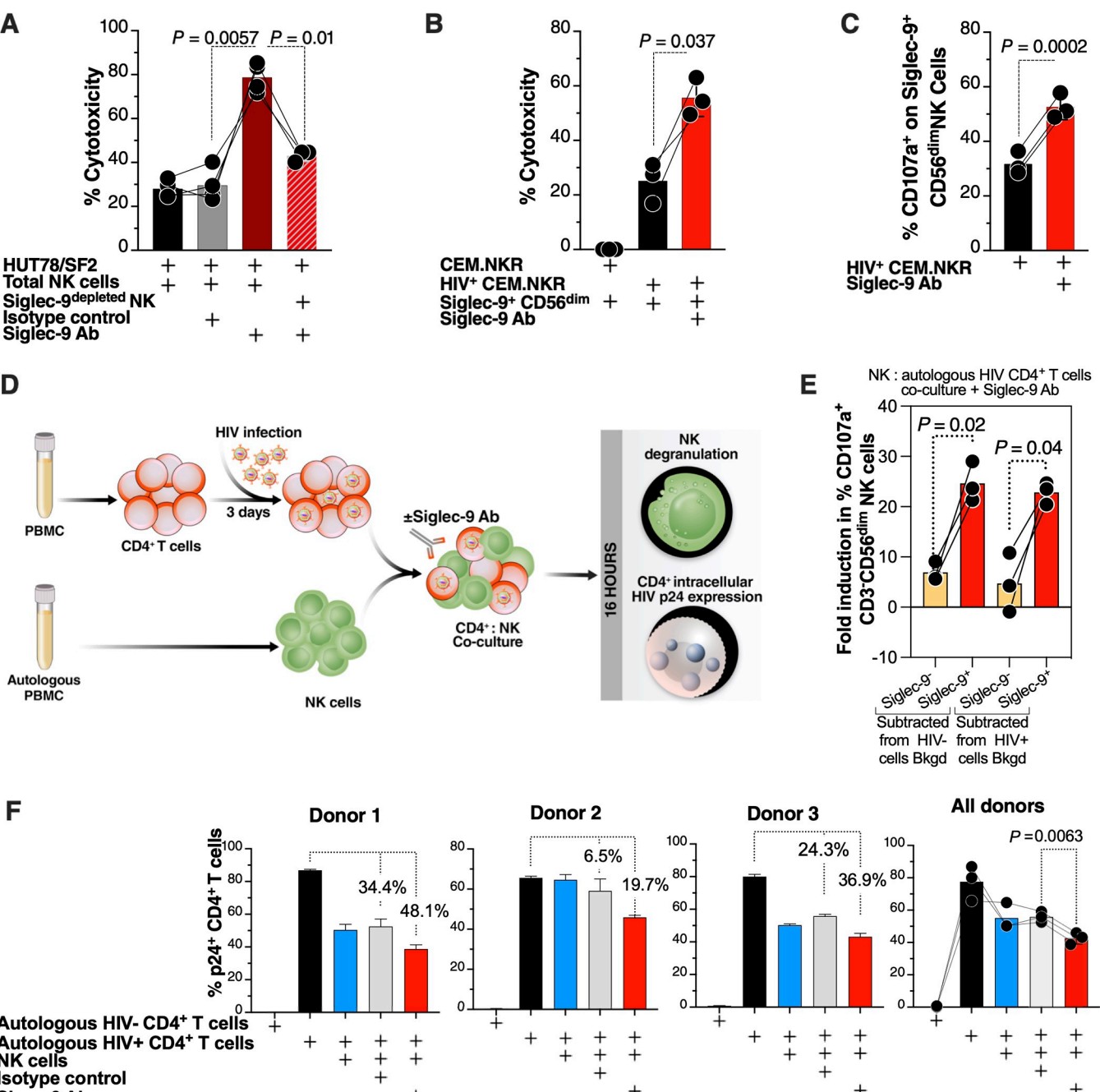

**Fig 5. The cytotoxicity of Siglec-9+ CD56dim NK cells towards HIV+ cells is being restrained by the inhibitory nature of the Siglec-9 molecule. (A)** HIV-infected HUT78/SF2 cells were used as targets, and total or Siglec9depleted NK cells from HIV-negative donors were used as effectors in the presence/absence of isotype control or Siglec-9 blocking Ab. Cytotoxicity was assessed by LDH release assay (E:T = 10:1). n = 4 donors (the last condition was performed on n = 3); assays from each donor were performed in 2–4 replicates, and the average of these replicates was used for statistical analysis using paired t-tests. **(B-C)** Blocking Siglec-9 enhanced the ability of Siglec-9+ CD56dim NK cells to target HIV-infected CEM.NKR cells. Cytotoxicity was assessed by **(B)** the LDH release assay (E:T = 10:1) or **(C)** NK degranulation (E:T = 4:1). Analysis of NK degranulation was made on total NK cells gated on Siglec-9+ or Siglec-9- CD56dim NK cell subsets. n = 3 donors; assays from each donor were performed in 2–4 replicates, and the average was used for statistical analysis using paired t-tests. **(D)** A schematic representation of the workflow to evaluate effector NK degranulation and cytotoxic potential against autologous HIV-infected CD4+ T cells in the presence or absence of Siglec-9 Ab. CD4+ T cells were isolated from fresh PBMC and exposed to HIV-1 IIIB for 72 h. On the third day, effector NK cells were isolated from PBMC of the same donor and co-cultured with autologous HIV-infected CD4+ T cells in the presence or absence of Siglec-9 Ab for 16 h. Both NK degranulation and intracellular p24 expression were evaluated by flow cytometry. **(E)** NK degranulation (CD107a expression) from the experiment described in (D). Assay from each donor was performed in triplicate (E:T = 2.5:1; n = 3 donors). Bkgd = background. Statistical analysis was performed using Paired t-tests. **(F)** Intracellular p24 expression from the experiment described in (D). Assay from each donor was performed in triplicate wells (E:T = 10:1; n = 3

donors). Percentages are percent reduction from the HIV-infected cells only condition. Statistical analysis was performed using paired ANOVA with post-hoc Holm-Sidak method (to correct for multiple comparisons).

### HIV bNAb-sialidase conjugates selectively desialylated HIV-infected cells

We next tested whether the HIV bNAb-sialidase conjugates can selectively remove sialic acid from HIV-infected cells. A mixture of HUT78/SF2 (HIV+ cells) and HUT78 (HIV-negative cells) was treated with each of the three bNAb-sialidase conjugates at escalating doses. Cells were then stained with a secondary antibody for anti-HIV antibody and SNA (a lectin that binds specifically to Sialic acid). Treatment with the NIH45-56-STSia conjugate selectively desialylated HIV+ cells, while HIV-negative cells were minimally affected (**Fig 6B and 6C**). Similar results were obtained with the 3BNC117-STSia (**Fig 6D**) and PGT151-STSia (**Fig 6E**) conjugates.

### HIV bNAb-sialidase conjugates enhance NK cytotoxicity against HIV-infected cell line

Next, we tested whether these conjugates can enhance NK-mediated cytotoxicity against HIV-infected cells. As targets, we utilized CEM.NKR CCR5$^+$ Luc$^+$ cells as these are 1) infectable with HIV (**S9A Fig**); 2) naturally resistant to NK killing without HIV infection; 3) express luciferase as a marker of HIV infection; and 4) can be desialylated by STSia to remove Siglec-9 ligands from their cell-surface (**S9B Fig**) which enhances NK activity against them (**S9C Fig**). We tested the ability of each of the three conjugates to potentiate the killing of HIV IIIB-infected cells by primary NK cells isolated from HIV-uninfected donors. NK cells alone were able to reduce infected cells by 2.5-fold. The addition of 50 nM NIH45-46 enhanced this to 7.3-fold. The addition of 50 nM NIH45-46-sialidase conjugate enhanced the reduction by 42.4 fold ($P = 0.0002$; **Fig 7A**) to levels almost matching that of uninfected cultures. Similar results were obtained using 200 nM of the antibody or antibody-sialidase conjugate (**Fig 7A**) and upon using 3BNC117 or PGT151 conjugates (**Fig 7B and 7C**). Cytotoxicity as measured by levels of lactate dehydrogenase (LDH) released into the supernatant from damaged cells also yielded similar results—the bNAb-STSia conjugates enhanced the ability of NK cells to kill HIV-infected cells compared to the unconjugated bNAbs (**Fig 7D–7F**).

To examine the direct role of the sialidase on the ability of bNAb-sialidase conjugate to enhance NK cytotoxic responses, we tested the effects of an Fc receptor blocker. Fc receptor blocker decreased the ability of bNAb to reduce levels of HIV-infected cells, as expected. However, the bNAb-sialidase conjugate was still able to enhance the ability of NK cells to reduce levels of HIV-infected cells (compared to the bNAb alone control), despite blocking Fc receptors (**Fig 7G**). These data suggest a direct role of the sialidase in the ability of the bNAb-sialidase conjugate to enhance NK cytotoxic responses.

We next examined the ability of NIH45-46-STSia to potentiate the cytotoxicity of NK cells isolated HIV-infected ART-suppressed individuals against CEM.NKR eGFP$^+$ cells (green) infected with HIV. Uninfected CEM.NKR CCR5$^+$ Luc$^+$ were labeled with PKH26 (red dye) and mixed with HIV-infected CEM.NKR eGFP$^+$. This cell mixture was then co-cultured with NK cells (isolated from donor ART09 in S1 Table) in the presence or absence of isotype control, NIH45-46, or NIH45-46STSia. This experiment was done in four replicates. Live cell imaging data shown in **Fig 7H and 7I** suggest that the NIH45-46-STSia conjugate enhances the ability of NK cells to eliminate HIV-infected cells while minimally impacting HIV-negative cells. This experiment was repeated using NK cells isolated from donor ART05 (S1 Table) with similar results (**S10 Fig**). Together, these data suggest that the bNAb-sialidase conjugates can selectively induce NK cytotoxicity against HIV-infected cell line compared to the bNAb alone.

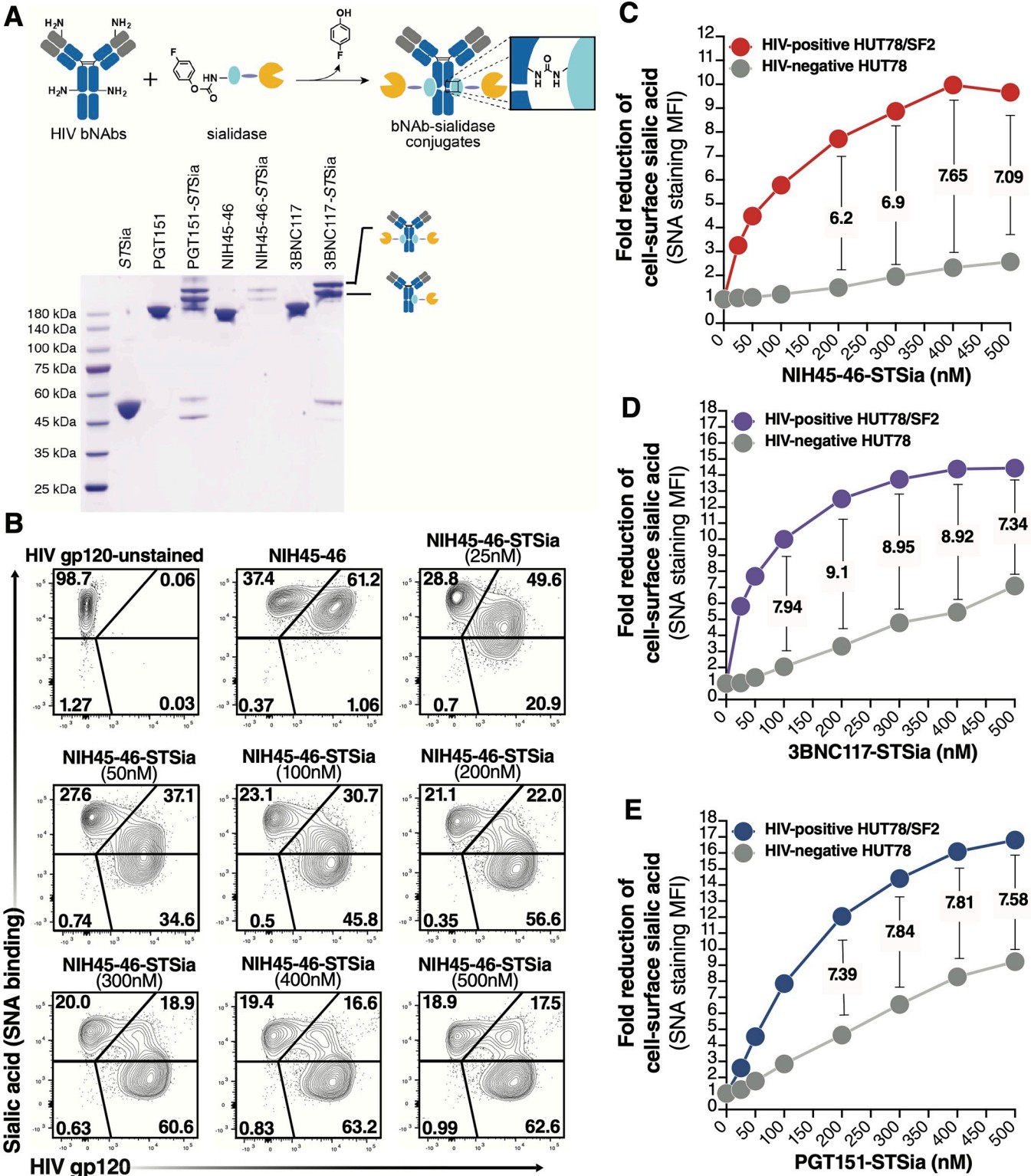

**Fig 6. bNAb-Sialidase conjugates selectively target HIV+ cells for desialylation. (A)** Top: preparation of site-specifically labeled HIV bNAb-Sialidase (bNAb-STSia) conjugates. Antibody-binding peptide (light blue) genetically fused with Sialidase (yellow) is conjugated to bNAb using pClick. pClick enables a site-specific conjugation between the antibody-binding peptide with payload and Lys337 of antibodies. Bottom: SDS-PAGE analysis with non-reducing buffer of bNAb-Sia conjugates. The two new bands above 180 kDa are consistent with the formation of the mono- or double-STSia antibody complex. **(B-C)** A mixture of HUT78 cells (HIV-negative) and HUT78/SF2 cells (HIV+) was treated with escalating doses of NIH45-46-STSia. HIV gp120 was measured by a

secondary antibody to NIH45-46, and Sialic acid levels were measured as binding to SNA lectin. Representative flow plots (**B**). The fold reduction shows that sialic acid was reduced by >7 fold on HIV$^+$ cells compared to HIV$^{-negative}$ cells (**C**). (**D-E**) Cells were treated as in B/C but using the 3BNC117-STSia conjugate (**D**) or the PGT151-STSia conjugate (**E**). STSia = Sialidase from *Salmonella typhimurium*.

### HIV bNAb-sialidase conjugates enhance NK cytotoxicity against autologous HIV-infected CD4$^+$ cells

Finally, we tested whether the bNAb-sialidase conjugates can enhance NK-mediated cytotoxicity against autologous HIV-infected primary CD4$^+$ T cells (infected with HIV-1 IIIB) (**Fig 8A**). In these experiments, we used HIV-infected primary CD4$^+$ T cells as target cells and autologous PBMCs as effector cells (**Fig 8A**). We first examined the impact of the NIH45-46-STSia conjugate on NK degranulation as measured by the frequency of CD107a$^+$, CD107a$^+$IFNγ$^+$, and CD107a$^+$TNFα$^+$ CD56$^{dim}$ NK cells (**Fig 8A**). These experiments were independently repeated three times using cells from three donors, and each repeat was performed in multiple replicates. A final concentration of 100nM or 300nM of NIH45-46-STSia significantly enhanced NK degranulation compared to 100nM or 300nM of NIH45-46, respectively (**Fig 8B–8E**). We next examined the impact of the NIH45-46-STSia conjugates on levels of HIV-infected cells, as measured by levels of intracellular HIV p24 (**Fig 9A**). Consistent with the NK degranulation data, 100 nM or 300 nM of NIH45-46-STSia significantly reduced levels of HIV-infected cells compared to 100 nM or 300 nM of NIH45-46, respectively (**Fig 9B–9E**). Together, these data suggest that disrupting Siglec/sialoglycan interactions by selectively desialylating HIV-infected cells, using antibody-Sialidase conjugates, may be an effective approach to enhance NK cell cytotoxicity against antigen-producing HIV-infected cells.

## Discussion

In this study, we identified the Siglec-9$^+$ CD56$^{dim}$ NK subpopulation, which has not been implicated during HIV infection, as a highly cytotoxic NK subpopulation against HIV-infected cells. We also found that the cytotoxicity of this subpopulation is restrained by the inhibitory nature of the Siglec-9 molecule itself. Harnessing the cytotoxic capacity of Siglec-9$^+$ CD56$^{dim}$ NK subpopulation, which is dampened by Siglec-9 expression, should be evaluated as a novel approach to control HIV infection during and/or after ART. Towards this goal, we developed a proof-of-concept approach to selectively disrupt the Siglec/sialoglycan interactions between NK cells and HIV-infected cells. Indeed, this approach showed specificity and efficacy in enhancing NK activity against HIV-infected cells *in vitro*.

The cytotoxic potential of NK cells is regulated through the balance of opposing signals delivered by inhibitory and activating cell surface receptors [45–47]. HIV infection induces phenotypic changes in NK cells and reduces their cytotoxicity [48], and some of these changes persist even after ART suppression of viral replication [49]. Among the emerging inhibitory receptors on NK cells are the Siglecs [20, 23–25,50,51]. Whereas the role of Siglec-7 in HIV pathogenies has been studied [17–19], to our knowledge, the role of Siglec-9$^+$ CD56$^{dim}$ NK cells in HIV infection has never been examined despite the growing appreciation of their role as glyco-immune negative checkpoints in cancer and HBV infection.

During both cancer and HBV infection, the frequency of cytotoxic Siglec-9$^+$ NK cells is reduced [24,25]. Our data suggest a similar reduction in the frequency of Siglec-9$^+$ CD56$^{dim}$ NK cells during HIV infection. However, there are several alternative explanations of this finding, including: 1) a change in the phenotype of these cells during HIV infection; 2) an increase in the frequency of other NK cell sub-populations; and 3) an increase in the migration ability

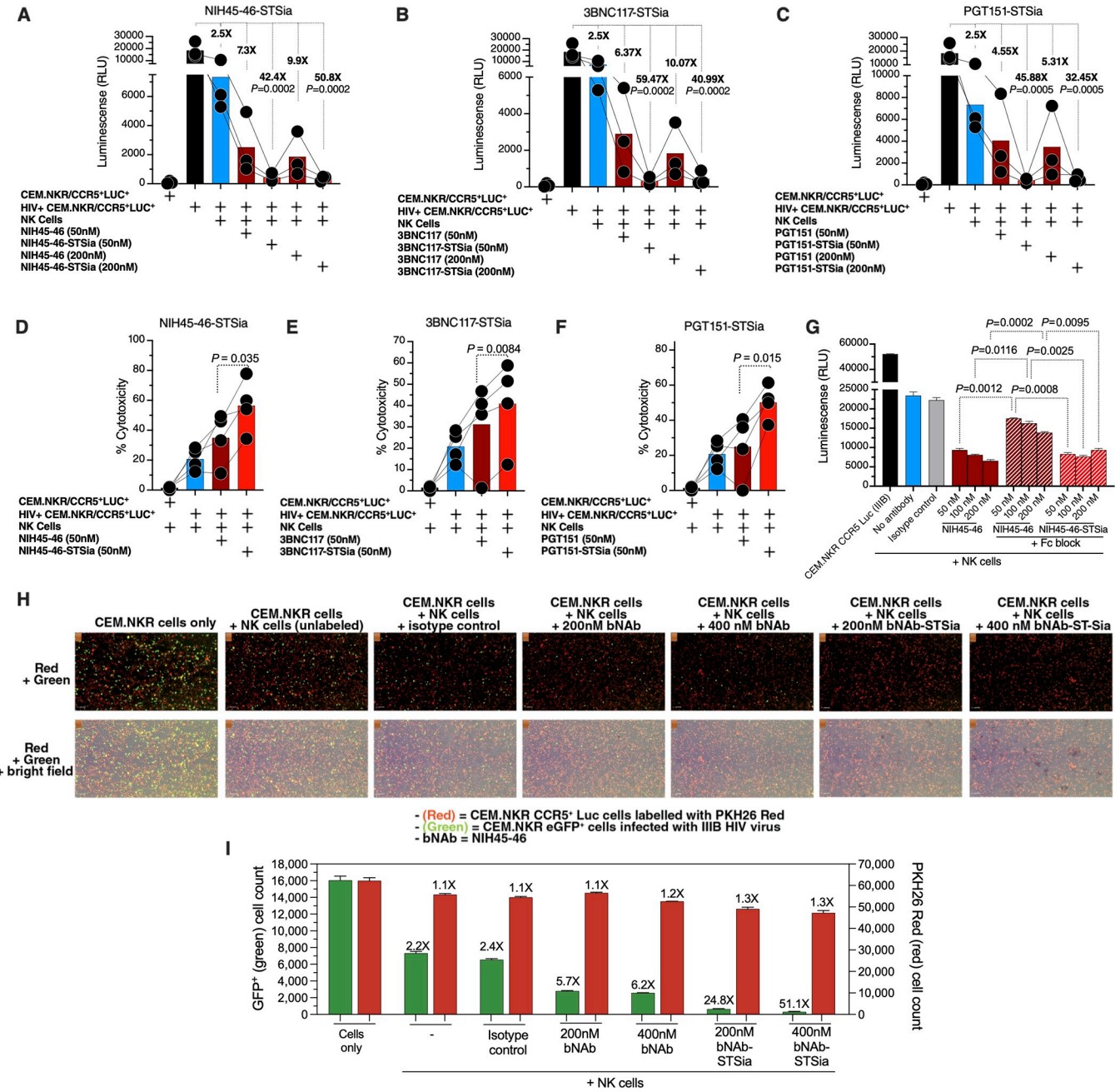

**Fig 7. bNAbs-STSia conjugates promote higher NK cytotoxicity against HIV+ cells compared to bNAbs alone. (A-C)** Killing assay using HIV-infected CEM-NKR CCR5+ Luc+ cells as targets and HIV-negative primary NK cells as effectors (n = 3 donors; assays from each donor were performed in 2–4 replicates, and the average of these replicates was used for analysis) at E:T ratio of 10:1. Luminescence was measured as a marker of intact (unkilled) HIV+ cells. **(A)** NIH45-46 and its conjugate. **(B)** 3BNC117 and its conjugate. **(C)** PGT151 and its conjugate. *P*-values were calculated using paired ANOVA with post-hoc Holm-Sidak method (comparing each condition against the HIV+ cells alone condition). **(D-F)** Killing assay using HIV-infected CEM-NKR CCR5+ Luc+ cells as targets and HIV-negative primary NK cells as effectors (n = 4 donors; assays from each donor were performed in 2–4 replicates, and the average of these replicates was used for analysis). Cytotoxicity was assessed by the LDH release assay at an E:T ratio of 10:1. **(D)** NIH45-46 and its conjugate. **(E)** 3BNC117 and its conjugate. **(F)** PGT151 and its conjugate. p-values were calculated using paired t-tests. **(G)** NK cells were treated with human Fc receptor blocking solution prior to co-incubation with HIV-infected CEM.NKR CCR5+ Luc+ cells. Luminescence was measured following 16 h incubation at a 10:1 E:T ratio. Unpaired ANOVA with post-hoc Dunnett T3 method (to correct for multiple comparisons) was used for statistical analysis between the indicated groups. **(H)** Effector NK cells were isolated from PBMC of an ART-suppressed HIV+ donor (ART09) and co-cultured with a mixture of HIV-uninfected PKH26-labeled CEM.NKR CCR5+ Luc+ (red cells) and HIV-infected CEM.NKR eGFP+ cells (green cells). Cell mixture was treated with NIH45-46, NIH45-46STSia, or isotype control. After 24 of co-culture, the Celigo image cytometer was used to directly visualize and count the number of PKH26-labeled (red) and GFP+ (green) target cells. The panel shows representative images from two independent experiments. This experiment was performed in quadruplicate at E:T 10:1. **(I)** Plot of the raw

GFP+ green (HIV-infected) cell count (left y-axis) and red PKH26-labeled (HIV-uninfected) cell counts (right y-axis) from (H). The fold reduction compares the average of each condition to the cell-only condition.

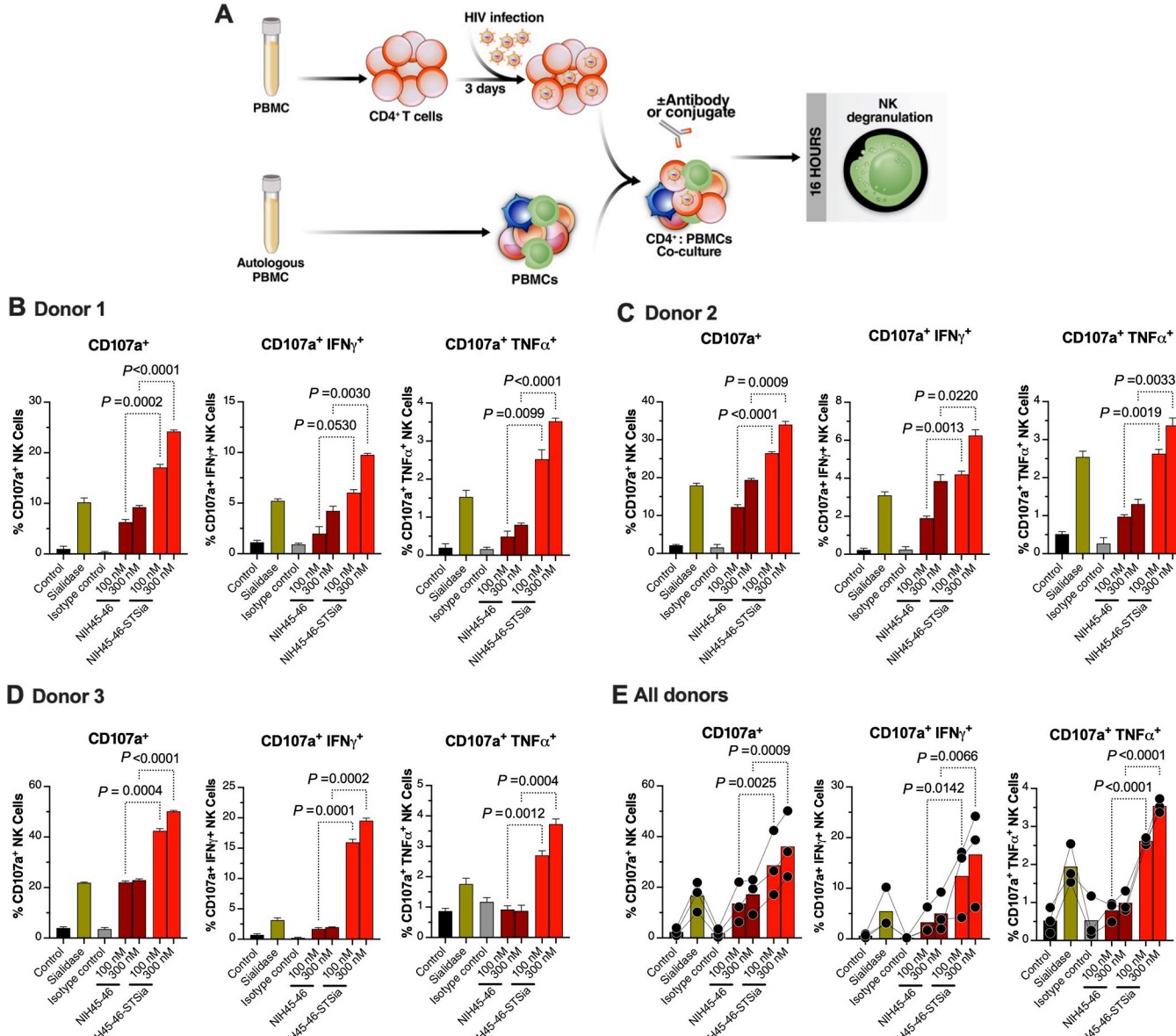

**Fig 8. NIH45-46-STSia induces NK degranulation towards autologous primary HIV-infected CD4+ T cells. (A)** A schematic representation of the workflow to evaluate effector NK degranulation against autologous HIV-infected CD4+ T cells in the presence of bNAb or bNAb-STSia conjugate. CD4+ T cells were isolated from fresh PBMC and exposed to HIV-1 IIIB for 72 h. On the third day, virus-infected CD4+ T cells were treated or not with Sialidase, NIH45-46, NIH45-46-STSia, or isotype-matched control antibody. PBMC from the same donor were co-cultured with autologous HIV-infected CD4+ T cells in the presence of CD107a antibody for 16 h. Following overnight incubation, the mixtures were stained with a cocktail of antibodies for CD3, CD56, IFN-γ, and TNF-α. Percent NK cell positive for CD107a, IFN-γ, and TNF-α expression was derived after gating on CD3-CD56dim NK cells. Control = data from the NK cells + HIV-infected CD4+ T cells condition. All other conditions contain NK cells + HIV-infected CD4+ T cells, in addition to the indicated reagent. **(B)** Data from donor 1. **(C)** Data from donor 2. **(D)** Data from donor 3. **(E)** Average data from all donors. Assay from each donor was performed in 4 replicate wells (E:T 10:1; n = 3 donors). Statistical analyses for panels B-D were performed using unpaired ANOVA with post-hoc Dunnett T3 method (to correct for multiple comparisons) comparing the indicated groups. Statistical analysis for panel E was performed using paired ANOVA with post-hoc Holm-Sidak method (comparing the indicated conditions).

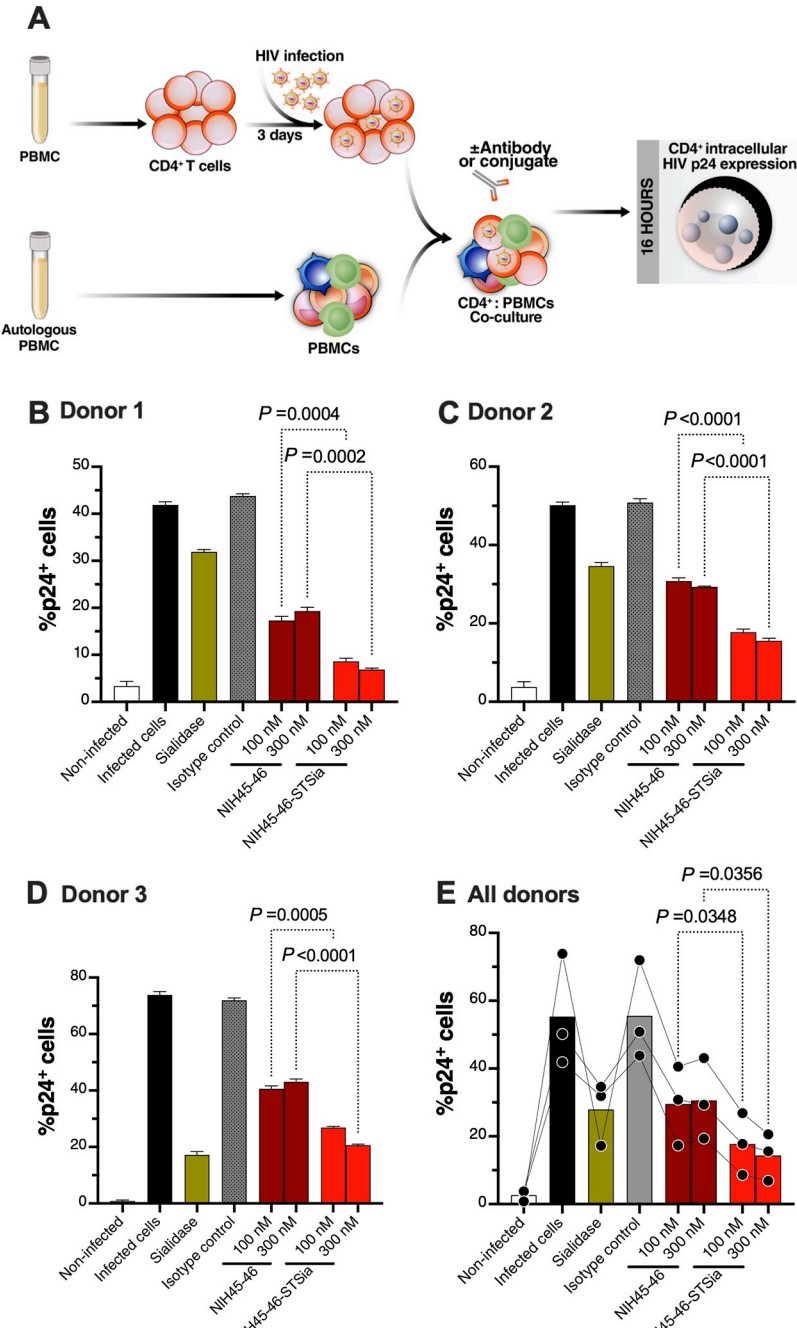

**Fig 9. NIH45-46-STSia induces PBMC cytotoxicity towards autologous primary HIV-infected CD4+ T cells. (A)** A schematic representation of the workflow to evaluate the cytotoxicity of PBMC against autologous HIV-infected CD4+ T cells in the presence of bNAb or bNAb-STSia conjugate. CD4+ T cells were isolated from fresh PBMC and exposed to HIV-1 IIIB for 72 h. On the third day, virus-infected CD4+ T cells were treated or not with Sialidase, NIH45-46, NIH45-46-STSia, or isotype-matched control antibody. PBMC from the same donor were co-cultured with autologous HIV-infected CD4+ T cells for 16 h. Following overnight incubation, the mixtures were stained for live/dead viability, CD3, CD8, and intracellular p24. Percent p24+ was derived after gating on CD3+, CD8- and live cells. **(B)** Data from donor 1. **(C)** Data from donor 2. **(D)** Data from donor 3. **(E)** Average data from all donors. Assay from each donor was performed in 4 replicate wells (E:T 100:1; n = 3 donors). Statistical analyses for panels B-D were performed using unpaired ANOVA with post-hoc Dunnett T3 method (to correct for multiple comparisons) comparing the indicated groups. Statistical analysis for panel E was performed using paired ANOVA with post-hoc Holm-Sidak method (comparing the indicated conditions).

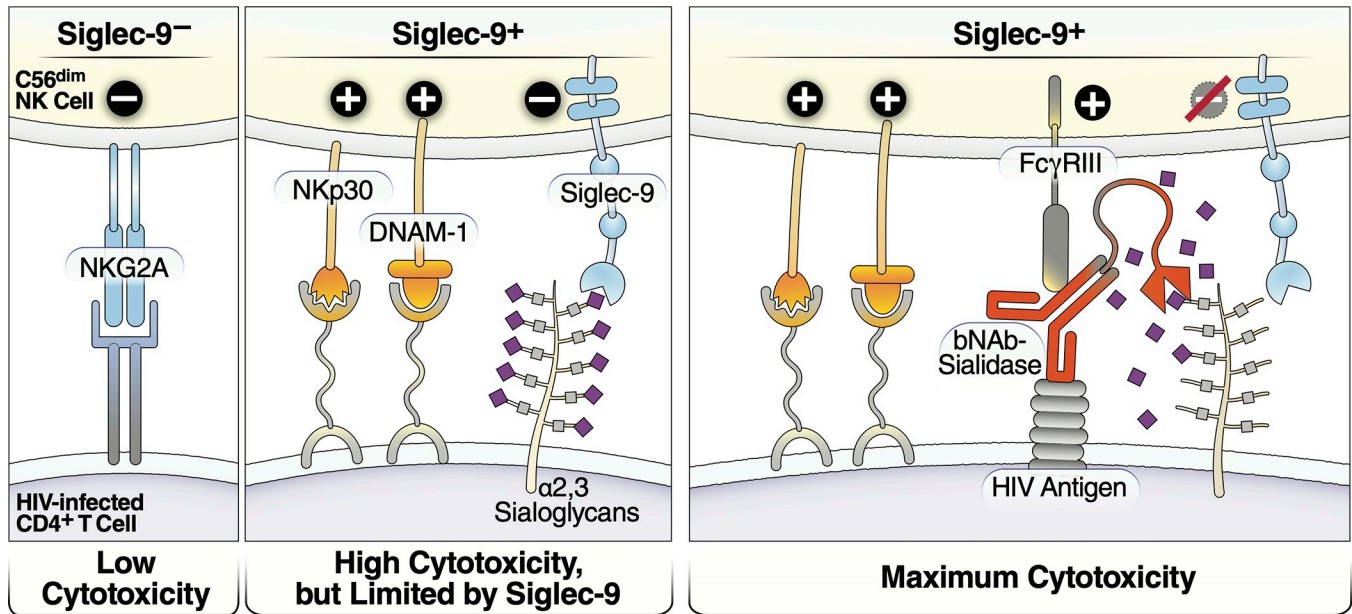

**Fig 10. Model of how HIV bNAb-Sialidase conjugates may increase the cytotoxicity of Siglec-9+ NK cells against HIV-infected cells.** Left two panels: The Siglec-9+ CD56dim NK subset has high cytolytic activity, possibly due to elevated expression of several NK activating receptors and reduced expression of the inhibitory NKG2A, compared to Siglec-9- CD56dim NK cells. However, Siglec-9 itself is an inhibitory receptor whose signaling restrains the cytolytic ability of these otherwise highly cytotoxic Siglec-9+ CD56dim NK cells by binding to Sialic acid attached to protein or lipid backbones on the surface of target cells. Right panel: Siglec/Sialic acid interactions are being pursued as an approach to enhance NK cell cytotoxicity against cancer using antibodies conjugated to Sialidase. We developed a similar proof-of-concept approach–conjugating Sialidase to HIV bNAbs–that could be used in conjunction with strategies that reactivate HIV latently-infected cells to enhance NK cells' capacity to clear HIV+ cells.

of these cells from the blood to other body compartments. In HBV-infected individuals, the percentage of Siglec-9+ NK cells inversely correlates with HBV DNA [25]. We found similar results in that HIV plasma viral load, during viremic HIV infection, and HIV CD4+ T cell-associated DNA levels, during ART-suppressed HIV infection, negatively associated with the levels of Siglec-9+ CD56dim NK cells. We confirmed that the Siglec-9+ CD56dim NK in HIV-infected individuals cells exhibit an activated phenotype with higher levels of activating receptors and markers (NKp30, CD38, CD16, DNAM-1, perforin) and lower expression of the inhibitory receptor NKG2A, compared to Siglec-9- CD56dim NK cells. Interestingly, while levels of Siglec-9+ CD56dim NK cells inversely correlated with levels of CD4+ T cell-associated HIV DNA during suppressive ART, Siglec-7+ CD56dim cells did not, hinting at a potentially distinct role of Siglec-9+ CD56dim NK cells. Our functional analysis demonstrated that sorted Siglec-9+ NK cells exhibit higher cytotoxicity towards HIV-infected cells compared to Siglec-9- NK cells, consistent with the highly cytotoxic nature of Siglec-9+ NK cells. However, blocking Siglec-9 further enhanced the ability of NK cells to kill HIV-infected cells. This result is consistent with the known inhibitory function of the Siglec-9 molecule itself on these otherwise cytotoxic cells. These data support a model in which Siglec-9+ CD56dim NK cells help in controlling HIV infection but are being restrained by the inhibitory nature of Siglec-9 receptor signaling (**Fig 10, left two panels**).

Little is known about Siglec-9 expression on NK cells in general, and no description of Siglec-9 phenotype or function on NK cells has been reported in the context of HIV infection. We performed both phenotypic and functional analyses of Siglec-9+ NK cells during HIV infection. Our data show that NK cells expressing Siglec-9 bear a more activated phenotype

(higher CD38) and a stronger functional profile (CD16, DNAM-1, Perforin, and NKp30) than Siglec-9⁻ NK cells. While this suggests that Siglec-9⁺ NK cells represent a more mature NK subset, we found no difference in CD57 expression between Siglec-9⁺ and Siglec-9⁻ NK cells in people living with HIV (PLWH) from our study. However, previous work from other groups showed that Siglec-9⁺ NK cells in control donors exhibit a more mature phenotype (higher CD57, CD69, and KLRG1) [24]. Previous studies have shown that Siglec-7⁺ CD56⁺ NK cells also have a stronger functional profile than Siglec-7⁻ CD56⁻ NK and Siglec-7⁻ CD56⁺ NK cells [17]. More recently, it was reported that the frequency of the CD11b⁺ CD57⁻ CD161⁺ Siglec-7⁺ subpopulation of CD56^dim CD16⁺ NK cells is lower in viremic HIV-infected individuals, correlates negatively with levels of HIV DNA, and exhibits higher cytokine and degranulation responses against K562 target cells [52]. Our data show that Siglec-7 is expressed at higher levels on Siglec-9⁺ CD56^dim NK cells. Future studies are needed to perform an in-depth phenotypic analysis of the Siglec-9⁺ CD56^dim NK cells subpopulation and determine its potential overlap with the CD11b⁺ CD57⁻ CD161⁺ Siglec-7⁺ subpopulation. Other NK subpopulations have also been identified as representing a mature and differentiated phenotype during HIV/SIV infection, such as the CXCR5⁺ [53] and NKG2A^Low [54] NK cells in secondary lymphoid tissues of SIV-infected African green monkeys (AGM). Future work will also need to address whether Siglec-9 is differentially expressed on these subpopulations in blood and in tissues, the main site for HIV persistence.

Our data highlight the Siglec-9/Sialic acid axis as a novel glyco-immune checkpoint mechanism that may be exploited by HIV-infected cells to evade immune surveillance by the cytotoxic Siglec-9⁺ NK cells. Understanding the potential role that Siglec-9/Sialic acid interactions play in the ability of HIV persistently-infected cells to evade NK immune surveillance during ART warrants further investigation. Future studies will need to determine the role of Siglec-9 in the ability of NK cells to kill HIV persistently-infected cells (latent and transcriptionally active cells that persist despite ART). These studies will also need to assess the role of cell surface expression of the Siglec-9 ligand, α2–3 Sialic acid, in the ability of these HIV persistently-infected CD4⁺ T cells to evade killing by NK cells. Here we focused on NK cells, but Siglecs are also expressed by myeloid cells and some T cells [20,55,56]. The interactions between Siglecs expressed on these other immune cells and HIV-infected cells warrant further investigation in future studies. Understanding these glycan-lectin interactions may allow for developing novel glycan-based tools to enhance immune functions during HIV infection to either cure HIV or prevent HIV-associated immune dysfunction.

Siglec/Sialic acid interactions are being pursued as an approach to enhance NK cell cytotoxicity against cancer using antibodies conjugated to Sialidase. We developed a similar proof-of-concept approach–conjugating Sialidase to HIV bNAbs–to augment NK killing activity towards HIV+ cells. HIV bNAbs are emerging as promising tools to cure HIV, but new tactics are needed to enhance their functionality [57,58]. We hypothesized that by conjugating Sialidase to bNAbs, we could enhance NK and bNAb functions in three ways: 1) Sialidase would be selectively steered to HIV+ cells by the bNAbs and desialylation of HIV+ cells would prevent Siglec/Sialic acid-binding (Siglec-7 and Siglec-9), which would enhance NK cytotoxicity; 2) NK function would be enhanced indirectly, as bNAbs would activate FCRγIII receptors (on NK cells), leading to antibody-dependent cell-mediated cytotoxicity (ADCC) of HIV+ cells; 3) for the bNAb, conjugating it with Sialidase would allow it to perform an additional, immunostimulatory function, augmenting NK cell ADCC. Indeed, in proof-of-concept experiments, we found that conjugating Sialidase to three different HIV bNAbs (NIH45-46, 3BNC117, and PGT151) selectively desialylated HIV-infected cells and enhanced the ability of NK cells to target and kill these infected cells *in vitro*. Unlike bNAbs or STSia alone, bNAbs-STSia conjugates significantly and specifically enhanced NK cytotoxicity towards HIV-infected targets,

highlighting their potential immunotherapeutic value. Such an immunotherapeutic approach can be used, in the future, in conjunction with strategies to reactivate latent HIV-infected cells, such as latency-reversing agents or ART-cessation. Desialylation of reactivated HIV-infected cells would further unleash the ability of Siglec-9-expressing NK cells to target and kill these cells to ultimately achieve a functional cure for HIV (**Fig 10, right panel**).

In this study, we used Sialidase from *Salmonella typhimurium* to conjugate the HIV bNAbs. The recently published cancer-focused data [40] using the same Sialidase to enhance NK functions against breast-cancer cells (in mice) suggest that this Sialidase is safe and efficient *in vivo*. However, other sialidases (including human sialidases (neuraminidases)) can and should be explored in the future. It is also important to note that Siglecs on NK cells bind to sialic acid on target cells *in trans* but also can bind to Sialic acid on NK cells themselves (*in cis*) [39,59]. Reassuringly, however, the recent cancer-focused data [40] using the same approach suggests that this approach has low off-target effects. While our study performed proof-of-concept *in vitro* examination of this novel approach, future studies should implement 1) *ex vivo* assays using cells from HIV-infected ART-suppressed individuals (in concert with latency-reversal agents); and 2) assessment of NK cell-mediated elimination of the reservoir *in vivo* using animal models. Within these studies, it could be explored to combine these bNAbs-sialidase conjugates with other approaches to cure HIV. For example, one could envision combining bNAbs-sialidase conjugates with shock-and-kill using efficient LRAs such as the recently described SMAC mimetics [60], or administering the conjugate during ART interruption to clear reactivated HIV+ cells before a full viral rebound.

Our study has limitations. First, regarding the *in vivo* examination in Figs 1–3, it will be important to examine the frequency and phenotype of Siglec-9$^+$ CD56$^{dim}$ cells in tissues, given that tissues are the main site for HIV persistence [7]. In addition, most of the cell-associated HIV DNA copies in HIV-infected ART-treated individuals harbor mutations and/or deletions, rendering them defective [61,62]. Examining the relationship between Siglec-9$^+$ CD56$^{dim}$ cells and levels of intact and inducible HIV DNA during ART will be needed. Next, for the NK cytotoxicity assays, we used certain viral isolates. In the future, it will be important to test other viral isolates, including transmitted/founder viruses. Furthermore, for the majority of these NK cytotoxicity assays, we used NK cells isolated from healthy donors (against HIV-infected cell lines or HIV-infected autologous CD4$^+$ T cells). However, validating our findings using cells from HIV-infected ART-suppressed individuals (from different clinical and demographic settings) will be important to ensure that HIV-mediated dysregulation of NK cell functions and phenotypes does not impact our findings. Regarding the experiments in Figs 7–9, we used the bNAbs alone as controls to examine the impact of bNAb-sialidase conjugates on NK cytotoxicity; it will be important, in the future, to examine bNAbs conjugated to non-specific enzymes. Finally, the proof-of-concept experiments in Figs 6–9 were exploratory in nature, and examining the utility of the novel conjugates we developed *in vivo* is needed. Despite these shortcomings, our study is the first to describe Siglec-9$^+$ CD56$^{dim}$ NK cells *in vivo* as an NK subpopulation that can be exploited by HIV-infected cells to evade immunosurveillance. Our study is also the first to describe that this population is highly cytotoxic but is being restrained by the inhibitory marker (Siglec-9) they express (analogous to PD1 expression in highly activated CD8$^+$ T cells). Finally, we developed a proof-of-concept approach (bNAb-sialidase conjugates) to selectively disrupt Siglec-9/sialoglycan interactions between NK cells and HIV-infected cells. This approach represents a promising, novel immunotherapeutic tool to be used in the future (in concert with latency-reversal agents or during ART-cessation) to clear reactivated HIV latently-infected cells.

## Materials and methods

### Ethics statement

Research protocols were approved by The Wistar Institute committee on Human Research (IRB# 2110176-6a). Written informed consent was obtained, and all data and specimens were coded to protect confidentiality. All human experimentation was conducted in accordance with the guidelines of the US Department of Health and Human Services and those of the authors' institutions.

### Human primary peripheral blood mononuclear cells (PBMCs)

We phenotypically characterized Siglec-9+ CD56$^{dim}$ NK cells from 31 donors (Figs 1 and 2): 10 HIV-negative controls; 11 HIV+ viremic; and 10 HIV+ on suppressive ART (clinical data of this cohort are in **S1 Table**). Frozen PBMCs of the HIV-infected ART-suppressed individuals were obtained from The Wistar Institute and the Philadelphia FIGHT cohort. Frozen PBMCs from HIV-negative and HIV-infected viremic individuals were obtained from the University of Pennsylvania Human Immunology Core, INER-CIENI (Mexico), and University of Toronto. We examined the relationship between Siglec-9 expression on CD56$^{dim}$ NK cells and levels of HIV DNA (Fig 3B) using cells from 11 HIV-infected ART-suppressed individuals (clinical data of this cohort are in **S2 Table**) obtained from the University of Hawaii at Manoa.

### Phenotypic characterization of Siglec-9+ CD56$^{dim}$ NK cells in Figs 1 and 2

Phenotyping of NK cells expressing Siglec-9 was performed on cryopreserved PBMC from HIV-, HIV+ ART+, and HIV+ viremic individuals as previously described [63]. In brief, cryopreserved PBMC were thawed and rested at 2 x 10^6 cells/ml for 3 hours (h) in a complete R10 medium (RPMI 1640 supplemented with 10% FBS, 1% L-glutamine, and 1% penicillin/ streptomycin) with 1 μl/ml of DNAse I (Roche, Branchburg, NJ) in the incubator at 37˚C, 5% $CO_2$. After resting, 2 x 10^6 PBMC/well were then plated into a 96 V-bottom well plate for staining. All staining steps were performed at room temperature in the dark. Cells were washed with PBS and resuspended in 45 μl/well of PBS. For viability exclusion, 5 μl/well of a 1:60 dilution of Live/Dead Fixable Aqua Dead Cell Stain Kit (Invitrogen) was added and incubated for 10 minutes. The extracellular antibody cocktail was then added in a volume of 50 μl/ well prepared in 1:1 solution of FACS buffer (0.1% sodium azide and 1% bovine serum albumin in 1X PBS) and BD Brilliant Stain buffer (BD Biosciences), reaching a final staining volume of 100 μl/well. The extracellular stain was incubated for 20 minutes. Cells were washed with FACS buffer and then fixed and permeabilized using the Foxp3/Transcription Factor Staining Buffer Set (eBioscience) following the manufacturer's instructions. The intracellular antibody cocktail was prepared in a final volume of 100 μl/well diluted 1X perm/wash buffer, added, and incubated for one hour. Finally, cells were washed with 1X perm/wash buffer and fixed with 1% paraformaldehyde. Cells were stored at 4˚C in the dark until acquisition on a BD FACS Symphony A5 flow cytometer (BD Biosciences) within 24 h. All flow cytometry data were analyzed using FlowJo V10.7.1 software (Tree Star, Ashland, OR).

The following antibodies were used for staining, from BD Biosciences: TIGIT Brilliant Blue 700 (clone 741182), Ki-67 Alexa Fluor 700 (clone B56), HLA-DR Brilliant Ultra Violet 395 (clone G46-6), CD8 Brilliant Ultra Violet 496 (clone RPA-T8), CD16 Brilliant Ultra Violet 615 (clone 3G8), CD38 Brilliant Ultra Violet 661 (clone HIT2), CD25 Brilliant Ultra Violet 737 (clone DX12), CD3 Brilliant Ultra Violet 805 (clone UCHT1), NKP30 Brilliant Violet 480 (clone P30-15), KIR3DL1 Brilliant Violet 711 (clone DX9), Granzyme B PE-CF594 (clone GB11) and CD161 PE-Cy5 (clone DX12). From Biolegend: Siglec-9 APC (clone K8), NKG2D

APC-Cy7 (clone 1D11), Perforin Brilliant Violet 421 (clone B-D48), CD14 Brilliant Violet 510 (clone M5E2), CD19 Brilliant Violet 510 (clone HIB19), CD56 Brilliant Violet 570 (clone HCD56), KLRG1 Brilliant Violet 605 (clone 2F1/KLRG1), NKP46 Brilliant Violet 650 (clone 9E2), PD-1 Brilliant Violet 750 (clone EH12.2H7), DNAM-1 Brilliant Violet 785 (clone 11A8), Siglec-7 PE (clone 6–434), CD57 PE Cy7 (clone HNK-1). KIR2DL2/S2/L3 B PE Cy5.5 (clone GL183) was obtained from Beckman Coulter and NKG2A Alexa Fluor 488 (clone 131411) from R&D Systems. Live/Dead Fixable Aqua Dead Cell Stain Kit (Invitrogen) was used for viability exclusion and was used following the manufacturer's instructions.

### CD7 and Siglec-9 staining of NK cells (S3 Fig)

Fresh human PBMCs from an HIV-negative donor were stained with a cocktail of anti-CD3 BV421 (clone UCHT1 BD), anti-CD56 PerCP Cy5.5 (clone B159 BD), anti-CD7 FITC (clone CD7-6B7 eBioscience), anti-Siglec-9 PE (clone K8 BioLegend), and PE-conjugated isotype-matched control antibody (clone MOPC-21 BioLegend) for 15 min at room temperature. Cells were washed twice, fixed, and analyzed by flow cytometry (LSR II, BD).

### Phenotypic characterization of Siglec-9$^+$ CD56$^{dim}$ NK cells in Fig 3B

Cryopreserved PBMC were thawed in warm 10% cRPMI (RPMI 1640 medium; (Hyclone, Logan, Utah) supplemented with 10% fetal bovine serum (FBS) (Hyclone), 1% penicillin-streptomycin (Hyclone), 10 mM HEPES (Hyclone), 2 mM L-glutamine (Hyclone), and 10 µg/ml DNase I (Sigma-Aldrich, Dorset, United Kingdom), washed with PBS + 2% FBS (Hyclone) (complete RPMI). Cells were stained for viability with an aqua amine reactive dye (AARD; Invitrogen), then incubated with panels of conjugated anti-human monoclonal antibodies: CD3-ECD (clone UCHT1, Beckman Coulter), CD4-AF700 (clone RPA-T4, BD biosciences), CD8-Qdot 605 (clone 3B5, Invitrogen), CD14- APC-Cy7 (clone MφP9, BD biosciences), CD16-BV421 (clone 3G8, Biolegend), CD56-PE-Cy7 (clone B159, BD biosciences), CD161-FITC (clone HP-3G10, Biolegend), PD1-PerCP-Cy 5.5 (clone NAT105, Biolegend), Siglec-7-PE (clone 6–434, Biolegend), and Siglec-9-APC (clone K8; Biolegend). Cells were then washed with PBS + 2% FBS and then fixed in 1% paraformaldehyde (PFA, Electron Microscopy Sciences) before acquiring on a custom four laser LSRFortessa flow cytometer (BD Biosciences). Data were analyzed using Flowjo Software version 9.5 (Treestar).

### qPCR quantification of HIV DNA in isolated CD4$^+$ T cells

CD4$^+$ T cells were isolated from the PBMCs of HIV-infected ART-suppressed individuals using the Human EasySep Human CD4$^+$ T Cell Isolation Kit (StemCell Technologies). Isolated cells were lysed in RLT Plus Buffer (Allprep isolation kit, Qiagen). Total DNA was extracted from the lysates using the Allprep DNA/RNA/miRNA Universal Kit (Qiagen). Total HIV DNA was quantified with a qPCR TaqMan assay using LTR-specific primers F522-43 (5' GCC TCA ATA AAG CTT GCC TTG A 3'; HXB2522–543) and R626-43 (5' GGG CGC CAC TGC TAG AGA 3'; 626–643) coupled with a FAM-BQ probe (5' CCA GAG TCA CAC AAC AGA CGG GCA CA 3' [64] using the StepOne Plus Real-Time PCR System (Applied Biosystems). Cell-associated HIV DNA copy number was determined using a reaction volume of 20 µl with 10 µl of 2x TaqMan Universal Master Mix II, including UNG (Applied Biosystems), 4 pmol of each primer, 4 pmol of the probe, and 5 µl of DNA. Cycling conditions were 50˚C for 2 min, 95˚C for 10 min, followed by 60 cycles of 95˚C for 15s and 59˚C for 1 min. External quantitation standards were prepared from DNA isolated from ACH-2 cells in a background of HIV-1 negative human cellular DNA, calibrated to the Virology Quality Assurance (VQA, NIH Division of AIDS) cellular DNA quantitation standards. Cell counts were determined by qPCR

using human genomic TERT (Applied Biosystems). Copy number was determined by extrapolation against a 7-point standard curve (1–10,000 copies) performed in triplicate.

## Cell culture

HUT78, HUT78/SF2, CEMx174, CEM.NKR, CEM.NKR eGFP and CEM.NKR CCR5[+] Luc[+] cells were obtained through the NIH HIV Reagent Program, Division of AIDS, NIAID, NIH and cultured in RPMI 1640 supplemented with heat-inactivated 10% fetal bovine serum (FBS), L-glutamine (2 mM), penicillin (50 U/ml), and streptomycin (50 mg/ml) in vented T-75 culture flasks (Corning). K562 (ATCC) cells were also maintained in the same medium. Cultures were maintained at 37˚C with 5% $CO_2$. Expi293F cells (Thermo Fisher) were maintained in Expi293 expression medium at 37˚C with 8% $CO_2$. TZM-bl cells were obtained through the NIH HIV Reagent Program, Division of AIDS, NIAID, NIH and maintained in Dulbecco's Modified Eagle's Medium supplemented with L-glutamine, sodium pyruvate, glucose and 10% heat-inactivated fetal bovine serum (FBS) in vented T-75 culture flasks.

## HIV infection of CEM.NKR and CEM.NKR CCR5[+] Luc[+] cells

2 ml solution of HIV-1 IIIB (873,464 $TCID_{50}$/ml) or DH12 (667,959 $TCID_{50}$/ml) grown in CEMx174 cells were added to RetroNectin precoated dish (Takara Bio) and incubated at 37˚C for 6 h. Following incubation, virus solution was removed from the dish and 5 x$10^5$ cells CEM.NKR or CEM.NKR CCR5[+] Luc[+] cells were added. After a 72 h incubation at 37˚C, cells were washed extensively and used for downstream assays. To evaluate HIV infection, 2 x $10^5$ cells were pelleted and resuspended in fixation buffer (BD Cytofix/Cytoperm) for 20 min at 4˚C. After fixation, cells were permeabilized (BD Perm/Wash Buffer) and stained with 2.5 μl of PE-conjugated anti-p24 Ab KC57 (Beckman Coulter) and analyzed by flow cytometry.

## Human NK cell isolation

NK cells were isolated from peripheral blood mononuclear cells (PBMC) obtained from healthy donors by immunomagnetic negative selection using the EasySep Human NK Cell Isolation Kit (STEMCELL Technologies) following the manufacturer's protocol.

## Depletion of Siglec-9[+] NK cells

2.5 ug human Siglec-9 biotinylated antibody (R&D Systems) was combined with 1 x $10^6$ purified NK cells and incubated for 10 min at 4˚C. Excess antibodies were removed by washing twice with PBS supplemented with 0.5% BSA. Cells were further incubated with 25 μl streptavidin MicroBeads (Miltenyi Biotec) for 15 min at 4˚C. Cells were washed once and loaded onto pre-equilibrated LS columns (Miltenyi Biotec) according to the manufacturer's instructions. After washing extensively, cells in the eluate fraction were pelleted and resuspended in a suitable medium for downstream assay.

## Sorting of Siglec-9[+] and siglec-9[-] NK cells

Isolated human NK cells were resuspended in PBS buffer supplemented with 0.5% bovine serum albumin and stained with anti-CD3 BV510 (BD), anti-CD56 PerCP Cy5.5 (BD), and anti-Siglec-9 PE (BioLegend) for 15 min at 4˚C. Cells were washed twice and sorted using the MoFlo Astrios EQ, cell sorter (Beckman Coulter). Sorted cells were gated on the CD3[-] CD56[dim] population.

## NK cytotoxicity assay by lactate dehydrogenase (LDH) release

Indicated target cells were resuspended in serum-free RPMI medium at a concentration of $2 \times 10^5$ cells/ml. $2 \times 10^4$ target cells were plated in a 96-well V-bottom plate (Corning) in 100 μl RPMI. Next, $2 \times 10^5$ isolated effector NK cells resuspended in RPMI were added to the targets. Cells were mixed, pelleted at 200*g* for 2 min, and incubated at 37˚C for 16 h. Following incubation, 10 μl of supernatant was diluted 1:10 in LDH storage buffer in a 96-well round-bottom microplate following the LDH-Glo Cytotoxicity Assay protocol (Promega). Samples were further diluted two-fold in the LDH detection reagent. Luminescence was recorded with a plate reader after a 60-minute incubation at room temperature. Percent cytotoxicity was calculated as ((experimental RLU–effector spontaneous control RLU–target cell spontaneous control RLU) / (Target cell maximum control RLU–target cell spontaneous control RLU)) X 100. In experiments that involved siglec-9 antibodies, purified human NK cells were pretreated with anti-Siglec-9 antibody or isotype-matched control antibody for 1 h at 37˚C in 96-well microplates before the addition of indicated target cells.

## CFSE/SYTOX Red NK direct cytotoxicity assay

In this method, target cells are tracked with fluorescent dyes to distinguish them from unlabeled effectors. $5 \times 10^5$ target cells in 500 μl RPMI were labeled with 2 μM of the green fluorescent dye carboxyfluorescein diacetate succinimidyl ester (CFSE) (Life Technologies) for 1 h at 37˚C. Following incubation, labeling reaction was quenched with 10 ml complete growth medium. Labeled target cells were resuspended in a complete growth medium at a concentration of $2 \times 10^5$ cells/ml. $2 \times 10^4$ target cells were plated in a 96-well V-bottom plate (Corning) in 100 μl complete growth medium. Next, $2 \times 10^5$ isolated effector NK cells resuspended in complete growth medium were added to the targets. Cells were mixed, pelleted at 200*g* for 2 min, and incubated at 37˚C for 16 h. Control wells were adjusted to equal volumes with complete growth medium. Following incubation, 50 μl SYTOX Red is added to wells for a final concentration of 5 nM. Cells are analyzed by flow cytometry. The FITC channel was used to capture CFSE[+] events and APC channel, SYTOX Red[+] events. Percentage target cell death was calculated as the ((FITC[+] APC[+] events) / (FITC[+] APC[-] events)) X 100.

## NK degranulation and cytokine production assay

Target cells were combined with purified human NK cells at indicated effector-to-target ratios in complete growth medium in the presence of GolgiStop (BD) and anti-CD107a PE antibody (BD). The cell mixture was pelleted at 200*g* for 2 min and incubated at 37˚C for 16 h. Post incubation, cells were stained for surface markers with anti-CD56 PerCP Cy5.5 (BD) and anti-CD3 BV510 (BD) and in certain experiments anti-Siglec-9 PE-Cy7 (clone K8). Cells were washed twice, fixed (BD Cytofix/Cytoperm), and permeabilized (BD Perm/Wash buffer). Following permeabilization, anti-IFN-gamma BV421 (BD) antibody was used for intracellular staining. All assays were acquired via flow cytometry with an LSR II. NK cells were defined as CD3[-] and CD56[+]. Data were reported as the percentage of cells positive for CD107a and/or IFN-gamma. In experiments that involved siglec-9 antibodies, purified human NK cells were pretreated with purified anti-Siglec-9 antibody or isotype-matched control antibody for 1 h at 37˚C in 96-well microplates before the addition of indicated target cells.

## Detection of cell surface Siglec-9 ligands

$1 \times 10^5$ cells were resuspended in 100 μl PBS supplemented with 0.5% bovine serum albumin 0.1% sodium azide. 1 μg recombinant human Siglec-9-Fc (R&D Systems) was added and

incubated for 1 h at room temperature. Following incubation, cells were washed twice with PBS supplemented with 0.5% bovine serum albumin 0.1% sodium azide and incubated with PE Fc-specific goat anti-human IgG (eBiosceince); 1:250 dilution, for 20 min at room temperature. Cells were further washed twice, fixed (BD Cytofix/Cytoperm), and acquired by flow cytometry (LSR II, BD).

## Siglec-9 ligand detection after desialylating cells

$1 \times 10^5$ cells were resuspended in 100 μl PBS supplemented with 0.5% bovine serum albumin 0.1% sodium azide. 100 nM sialidase was added and incubated for 1 h at 37˚C. To remove residual Sialidase, cells were washed twice by centrifugation $400g$ for 5 min. Next, cells were evaluated for cell surface Siglec-9 ligand content following protocol to detect cell surface siglec-9 ligands as described.

## Isolation of autologous primary human NK and CD4$^+$ T cells

Primary human CD4$^+$ T and NK cells were isolated from fresh PBMCs obtained from HIV-negative donors by immunomagnetic negative selection using the EasySep Human CD4$^+$ Cell Isolation Kit and the EasySep Human NK Cell Isolation Kit (STEMCELL Technologies), respectively, following manufacturer's protocols. Contemporaneous PBMC was washed and cultured at 37˚C in RPMI 1640 supplemented with heat-inactivated 10% fetal bovine serum (FBS), L-glutamine (2 mM), penicillin (50 U/ml), and streptomycin (50 mg/ml). Isolated NK cells were cultured using the same medium. CD4$^+$ T cells were cultured in medium supplemented with 30 U/ml IL-2 (PeproTech).

## Virus infection of autologous primary CD4$^+$ T cells

Primary CD4$^+$ T cells ($2.5 \times 10^6$ cells) were added concurrently with bead CD3/CD28 to Retro-Nectin precoated dish that has been exposed to a 1 ml solution of HIV-1 IIIB (873,464 TCID$_{50}$/ml) and incubated at 37˚C for 16 h. Following overnight incubation, cells were removed from the dish and cultured in 24 well plates in RPMI 1640 supplemented with heat-inactivated 10% fetal bovine serum (FBS), L-glutamine (2 mM), penicillin (50 U/ml), streptomycin (50 mg/ml), and 30 U/ml IL-2. After 72 h incubation at 37˚C, beads were separated from the cells before washing twice with medium without IL-2 and evaluated for intracellular p24 by staining with anti-p24 Ab KC57-RD1 (clone FH190-1-1) (Beckman Coulter).

## Autologous NK degranulation assay

Purified NK cells were cultured alone or co-cultured with autologous HIV-infected CD4$^+$ T cells in complete growth medium in the presence of GolgiStop (BD) and anti-CD107a PE (clone H4A3) antibody (BD) for 16 h. Following incubation, cells were stained with anti-CD3 BV510 (clone UCHT1) (BD), anti-CD56 PerCP Cy5.5 (clone B159) (BD), anti-IFN-γ BV421 (clone B27) (BD) and anti-Siglec-9 PE-Cy7 (clone K8) antibodies. Analysis of Siglec-9$^+$ and Siglec-9$^-$ NK cell subsets were made on CD3$^-$ CD56$^{dim}$ gated cells. For experiments that involve antibody-sialidase conjugates, PBMC were cultured alone or co-cultured with autologous HIV-infected CD4$^+$ T cells that had been treated with Sialidase only (300 nM), isotype-matched antibody (300 nM) and indicated amount of antibody or antibody-sialidase conjugate for 2 h at 37˚C. Cells were mixed in the complete growth medium in the presence of GolgiStop (BD) and anti-CD107a PE antibody (BD) for 16 h. Following incubation, cells were stained with anti-CD3 BV510 (BD), anti-CD56 PerCP Cy5.5 (BD), anti-IFN-γ BV421 (BD) and anti-TNF-α (Clone MAb11) (BioLegend). NK cells were defined as CD3$^-$ and CD56$^+$. Background

NK degranulation (PBMC only control) was subtracted. Analysis of CD107a$^+$, IFN-$\gamma^+$, and TNF-$\alpha^+$ NK cells was made on CD3$^-$ CD56$^{dim}$ gated cells.

## Autologous NK cytotoxicity assay

2 x 10$^5$ FACS-sorted Siglec-9$^+$ and Siglec-9$^-$ NK cells were cocultured with autologous HIV-infected CD4$^+$ T cells (2 x 10$^4$ cells/well) for 16 h at 37˚C. Following incubation, cells were harvested and evaluated for intracellular p24 by staining with Zombie NIR fixable viability dye (BioLegend), anti-CD3 BV421 (clone UCHT1) (BioLegend), and anti-p24 KC57-RD1 (Beckman Coulter). For Siglec-9 blocking experiments, 2 x 10$^5$ purified NK cells in 100 μl medium were treated with anti-Siglec-9 antibody (100 μg/ml) or isotype-matched control antibody for 30 min at 37˚C in 96-well microplates before co-culture with autologous HIV-infected CD4$^+$ T cells (2 x 10$^4$ cells/well) for 16 h at 37˚C.

## Autologous PBMC cytotoxicity assay

HIV-infected CD4$^+$ T cells were treated with Sialidase only (300 nM), isotype-matched antibody (300 nM), and indicated amount of antibody or antibody-sialidase conjugate for 2 h at 37˚C. Autologous PBMC were co-cultured with virus-infected CD4$^+$ T cells in the complete growth medium for 16 h at 37˚C. Following incubation, cells were harvested and evaluated for intracellular p24 by staining with Zombie NIR fixable viability dye (BioLegend), anti-CD3 BV421 (clone UCHT1) (BioLegend), anti-CD8 FITC (clone HIT8α) (BD), and anti-p24 KC57-RD1 (clone FH190-1-1) (Beckman Coulter).

## Supernatant virus infectivity assay

The virus titers released into the supernatant from indicated co-culture conditions were evaluated for infectivity using TZM-bl cells, an indicator cell line that expresses HIV receptors and has luciferase genes under the control of the HIV-1 promoter. Supernatants, obtained after cells were pelleted, were serially diluted in a 96-well flat-bottom culture plate containing complete TZM-bl growth medium. 2 x 10$^4$ TZM-bl cells were then added to each well in the presence of 30 μg/ml DEAE-dextran. Following a 48h incubation, 100 μl of culture medium was removed from each well and replaced with 100 μl Bright-Glo luciferase substrate reagent (Promega). After 2 min, the well contents were mixed, and 150 μl of cell lysate were transferred to a clear-bottom black 96-well microplate. Luminescence (RLU) measurements were integrated over 0.1 s per well. Raw RLU values are shown.

## Generation and characterization of human Siglec-9 blocking antibody

Female BALB/c mice (5–7 weeks of age) were intramuscularly injected with 50 μg of synthetic DNA encoding human Siglec-9. Mice received two booster injections at two-week intervals, the first booster contained Siglec-9 DNA and the second booster contained 50μg of purified recombinant human Siglec-9 protein (R&D Systems). Mice spleens were then harvested and fused with SP2/0 mouse myeloma cells using the HY Hybridoma Cloning Kit (Stem Cell Technologies). Hybridomas were screened by ELISA for the presence of IgG antibodies directed against the human Siglec-9 antigen. In brief, soluble recombinant human Siglec-9 (R&D Systems) or His-Tag protein in PBS 1μg/mL (100μl/well) protein were coated overnight in 50mM sodium borate buffer (pH 8.0) at 4˚C. Following incubation, plates were washed with PBS-T (PBS with 0.05% Tween 20) and blocked using PBS containing 10% FBS. The plates were then washed three times with PBS-T, and incubated with various dilutions of hybridoma supernatants containing Siglec 9 antibody were added to the blocked plates and incubated at room

temperature for 1h. After another wash, the plates were treated with secondary antibody:goat-anti-mouse Peroxidase-AffiniPure Goat anti-Mouse IgG at a dilution of 1:10000 for 1 h at room temperature. Post a final wash, the plates were developed with OPD substrate (Sigma-Aldrich) for 5–10 min in the dark, and the reaction was stopped using 1N $H_2SO_4$. The plates were read using a Synergy2 plate reader (BioTek Instruments) at an optical density of 450nm. Approximately 1152 antibody-producing mouse hybridoma clones were identified as having antibody binding at least three-fold higher than the background level reactivity, were sub-cloned to generate hybridoma clones [65]. Selected clones were tested for their ability to enhance NK functions against the K562 cell line. Selected clone from this analysis was sequenced, and recombinant antibody heavy and light chain sequences were assembled into IgG1 framework and cloned into pcDNA3.4 antibody expression vectors, as previously described [65]. Plasmids were then transfected into Expi293F cells using the Expifectamine 293 Expression Kit (Thermo Fisher Scientific), and recombinant Ab was purified with protein A agarose (Invitrogen) [65].

The binding of the recombinant Siglec-9 Ab to recombinant human Siglec-9 protein was measured by ELISA (**S11A Fig**), as described above. To determine the specificity of the recombinant anti-Siglec-9 Ab, western blot analysis was used. Human recombinant Siglec-9 (1μg/mL) and Siglec-3 (1μg/mL) proteins (R&D Systems) were reduced using NuPAGE Sample Reducing Agent (10x) (Thermo Fisher Scientific) and heating at 70˚C for 10 min, then loaded onto sample lanes with Odyssey Protein Molecule Weight (LI-COR) serving as a standard marker. The gel electrophoresis was carried out using sodium dodecyl sulfate-12% polyacrylamide gel for 50 min at 200V. Following electrophoresis, samples were transferred onto polyvinylidene fluoride (PVDF) membranes via an iBlot-2 system (Thermo Fisher Scientific) and blocked using Odyssey Blocking Buffer (OBB) (LI-COR) for 1–2 hours on a rocker. Membranes were treated with antibody culture supernatant (1:500) in OBB containing 0.1% Tween 20 overnight at 4˚C. Following incubation, the membranes were washed four times at 5 min intervals with PBS-T. Subsequently, washed membranes were treated with goat anti-mouse secondary antibody (IRDye 800 CW) in OBB containing 0.1% Tween 20 and 0.01% SDS at a dilution of 1:10000 and incubated for 60 minutes in the dark on the rocker at room temperature. Following incubation, the membranes were rewashed four times and scanned using Odyssey CLx Imager (LI-COR). Anti-Siglec-9 Ab showed binding specificity to recombinant Siglec-9 protein and did not bind to recombinant Siglec-3 protein (**S11B Fig**).

## HIV bNAbs expression and purification

To produce recombinant anti-gp120 antibodies, plasmids encoding codon-optimized heavy and light chains of PGT151, NIH45-46, and 3BNC-117 were commercially synthesized (GenScript) as previously described [41]. Expi293F cells were transfected with plasmid DNA encoding equimolar ratio of heavy and light chains of anti-gp120 antibodies using Expifectamine following the manufacturer's protocol (Thermo Scientific). 18 h after transfection, enhancers were added to the cultures. Antibody-containing supernatants were harvested four days post-transfection, clarified at 1,000*g* for 10 min at 4˚C. Supernatants were then filtered through a 0.45-μm filter unit (Fisher Scientific). To purify IgG from the supernatants, protein G Mag-Beads (GenScript) were then used following the manufacturer's protocol. Bound antibodies were separated using magnets and eluted with Pierce IgG Elution Buffer (Thermo Scientific). Eluted antibodies were filtered through a 0.2-μm filter unit. The concentrations of purified recombinant antibodies were determined using a NanoDrop by measuring absorbance at A280.

## Cell surface gp120 staining

Anti-gp120 antibody binding to HIV-infected cells was evaluated by flow cytometry. Indicated antibodies at 20 μg/mL were incubated with 2 x $10^5$ HUT78/SF2 cells in 100 μL PBS supplemented with 0.5% bovine serum albumin 0.1% sodium azide and incubated for 30 minutes at room temperature. After washing twice, cells were resuspended in 100 μL PE Fc-specific goat anti-human IgG (eBiosceince); 1:250 dilution, for 20 min at room temperature. Cells were again washed, fixed, and analyzed by flow cytometry.

## Expression and purification of FB-STSia protein

To construct pET151-NH-FB-STSia plasmid, FB-E25TAG gene was generated by PCR using primers CY060/JP001 and template pET-22b-T5-FB-E25TAG and inserted into pET151-STSia vector amplified with primers CY061/JP002. The pET22b-T5-NH-FB-STSia and pUltra-FPheKRS plasmids were co-transformed into E. coli DH10B strains. Cells were grown in LB medium, supplemented with ampicillin (50 ug/mL), spectinomycin (25 ug/mL) and 1 mM FPheK at 37˚C. When the OD reached 0.6, 1 mM IPTG was added to the culture, and the culture was grown over night at 30˚C. The cells were harvested by centrifugation at $4,700 \times$ g for 10 min. The cell pellets were suspended in lysis buffer and lysed at 30˚C. The resulting cell lysate was clarified by centrifugation at $14,000 \times$ g for 30 min, and the proteins were purified on Ni-NTA resin (Qiagen) following the manufacturer's instructions. Primer sequence: JP001 (FB-insert-f) 5'-atcatcaccatcaccatggtTCTGTGGACAATAAATTCAACAAGGAGCAACA-3'. CY060 (FB-insert-r) 5'- CGCACAAGCGCCTAAAGGATCCgaaaacctgtattttcagggaACGG-3'. CY061 (STSia-v-f) 5'- TGACGCACAAGCGCCTAAAGGATCCgaaaacctgtattttcagggaACGG TTGAAAAGTC-3'. JP002 (STSia-v-r) 5'- AGAaccatggtgatggtgatgat-3'. DNA sequence of FB-STsia protein: atgcatcatcaccatcaccatggtTCTGTGGACAATAAATTCAACAAGGAGCAA CAGAACGCCTTTTATGAAATCCTTCATTTGCCAAACCTTAACTAGGAACAGAGGAA TGCGTTTATTCAATCACTTAAGGATGATCCAAGTCAATCGGCCAACCTGTTGGCG GAGGCCAAAAAATTAAATGACGCACAAGCGCCTAAAGGATCCgaaaacctgtattttcaggga ACGGTTGAAAAGTCTGTCGTGTTTAAAGCCGAGGGTGAACACTTCACGGATCAAA AAGGCAATACTATTGTTGGGTCTGGGTCCGGAGGAACCACTAAATATTTCCGTAT CCCGGCGATGTGTACGACCAGTAAGGGAACGATTGTCGTTTTCGCCGATGCCCGT CACAACACCGCGAGCGACCAGAGTTTCATTGATACCGCGGCTGCGCGTTCCACTG ATGGAGGTAAAACTTGGAACAAAAAGATTGCCATCTACAATGACCGCGTGAATAG TAAGCTGTCACGCGTAATGGATCCTACTTGCATTGTTGCAAATATCCAAGGGCGC GAAACGATTCTGGTAATGGTAGGCAAGTGGAATAACAATGACAAGACCTGGGGTG CATATCGTGACAAAGCCCCAGACACCGACTGGGATCTGGTCCTTTACAAGAGCACC GACGATGGCGTTACTTTCAGCAAGGTGGAGACTAATATCCACGATATTGTCACAAA GAACGGCACGATCTCGGCTATGCTTGGCGGGGTCGGGTCGGGTCTTCAGCTTAAT GACGGCAAACTGGTCTTTCCGGTCCAAATGGTGCGTACTAAAAATATCACCACCGT CTTAAATACGAGCTTCATTTACTCTACAGATGGAATCACTTGGTCACTTCCCAGTG GGTATTGTGAGGGATTTGGTAGTGAAAACAACATCATTGAATTCAATGCCTCCTTA GTAAACAACATTCGTAACTCGGGTTTGCGTCGCAGCTTTGAAACGAAAGACTTCGG GAAAACGTGGACGGAGTTTCCTCCCATGGATAAGAAAGTGGACAATCGTAATCAC GGTGTGCAGGGTTCTACTATCACAATTCCGTCTGGGAACAAGCTTGTCGCAGCGCA TTCTTCAGCTCAGAACAAAAACAATGACTATACTCGTTCGGACATTAGTTTGTATG CTCACAATTTATACTCCGGTGAAGTGAAATTGATTGATGCTTTTTACCCAAAAGTC GGAAACGCGTCTGGGGCCGGGTACTCGTGTCTGTCTTATCGCAAAAATGTTGATAA AGAGACACTGTATGTAGTATATGAGGCTAATGGTAGTATTGAATTTCAAGATTTGT CTCGCCACCTTCCGGTGATCAAATCATATAATTGAtga.

## Proximity-induced antibody conjugation (pClick)–to conjugate HIV bNAbs to Sialidase

To site-specifically conjugate the HIV bNAbs with Sialidase (from *Salmonella typhimurium*) using pClick, we first genetically incorporated 4-fluorophenyl carbamate lysine into the Glu25 position of a FB fused with Sialidase using the genetic code expansion technology. Next, we prepared bNAb-Sia conjugates by incubating bNAbs with 16 equivalents of FB-Sia for 48 h. The resulting bNAb-Sia conjugates were then purified by size-exclusion column. We characterized the conjugates by SDS-PAGE and determined an enzyme/antibody ratio = 1.0.

## SDS-PAGE analysis of bNAbs-STSia conjugates

PGT151, NIH45-46, and 3BNC117 were incubated with eight equivalents of FB-STSia separately in PBS buffer (pH 8.5) at 37˚C for 48 h and purified by size exclusion chromatography. PGT151, NIH45-46, and 3BNC117 antibodies alone or after conjugation with the FB-STSia were resolved in non-reducing gel. The resulting gel was stained at room temperature and washed with deionized water.

## Antibody-STSia conjugate selectivity

An equal number of HUT78 and HUT78/SF2 cells were mixed and then treated with antibody-STSia conjugates or unconjugated anti-gp120 antibodies only for 1 h at 37˚C. Following incubation, cells were washed twice with PBS supplemented with 0.5% bovine serum albumin 0.1% sodium azide and costained with FITC-labeled-SNA and PE Fc-specific goat anti-human IgG (eBiosceince) for 30 min at room temperature. Cells were washed, fixed, and analyzed by flow cytometry. Cell surface sialylation levels were revealed with SNA-FITC, and gp120 levels were determined using anti-gp120-specific antibodies.

## CEM.NKR CCR5$^+$ Luc$^+$ ADCC assay

CEM.NKR CCR5$^+$ Luc$^+$ is a cell line derived from CEM.NKR CCR5+ that stably expresses the luciferase reporter gene under the transcriptional control of the HIV LTR. Upon HIV infection of these cells, *Tat* drives expression of luciferase, which can be quantified in the presence of a suitable substrate. HIV-infected and uninfected CEM.NKR CCR5$^+$ Luc$^+$ cells were washed with PBS and resuspended in RPMI. Cells were plated at $2 \times 10^4$ cells/well in a V-bottom microplate (Corning). HIV-infected CEM.NKR CCR5$^+$ Luc$^+$ were treated with indicated concentrations of anti-gp120 antibodies or Ab-ST-Sia conjugates for 2 h at 37˚C. Control wells without antibodies were adjusted to volume with RPMI. Following incubation, purified human NK cells were added to wells. Cells were mixed pelleted at 200$g$ for 2 min and incubated for 16 h at 37˚C. Following incubation, 10 μl of supernatants were subject to LDH analysis. To evaluate luciferase activity, 100 μl supernatant was removed from all wells and replaced with 100 μl Bright-Glo luciferase substrate reagent (Promega). After 2 min, the well contents were mixed and transferred to a clear-bottom black 96-well microplate. Luminescence (RLU) measurements were integrated over 1 second per well. Raw RLU values are shown relative to the light output generated in RPMI medium only (background).

## Fc receptor block assay

Prior to co-culture with virus-infected infected CEM.NKR CCR5$^+$ Luc$^+$ cells, purified human NK cells were treated for 15 min with 10 μl of the human TruStain FcX Fc receptor blocking solution (BioLegend). Examining the impact of the Fc receptor block on HIV infection in infected CEM.NKR CCR5$^+$ Luc$^+$ cells was done as described above.

## Image-based NK cell cytotoxicity

CEM.**N**KR eGFP$^+$ cells were exposed to HIV-1 for 72 h. On the third day, cells were washed extensively to remove free virus. CEM.NKR CCR5$^+$ Luc$^+$ cells were labeled with PKH26 dye (Sigma) following the manufacturer's instructions. Both cells were mixed and seeded into a V-bottom 96 well plate. The cell mixture was treated or not with NIH45-46-STSia, NIH45-46, or an isotype-match antibody at indicated concentrations for 2 h at 37°C. Effector NK cells were isolated from frozen PBMC of ART-suppressed HIV donors. 2x10$^5$ NK cells were co-cultured with the PKH26-labeled CEM.NKR CCR5$^+$ Luc$^+$ / CEM.NKR eGFP$^+$ (HIV) cell mixture for 24h. Control wells received no effector cells. Images were acquired after 24h using the Nexcelom Celigo image cytometer. The Celigo expression analysis application was used with target 1 + 2 + 3 (brightfield + green + red). The green channel was used to detect GFP$^+$ CEM.NKR eGFP$^+$ (HIV) cells; the red channel, the PKH26-labeled CEM.NKR CCR5$^+$ Luc$^+$ cells. The whole well cell counts were acquired and plotted. Representative field images are shown.

## Statistical analysis

Data were analyzed using Prism 9.0 (GraphPad Software). Mann-Whitney tests were used for the analyses in Figs 1D and 2C (between groups), and S2D Fig. Wilcoxon signed rank tests for paired data were used to compare between Siglec-9$^+$ and Siglec-9$^-$ within each group in Fig 2B and 2C. Spearman's rank correlation was used in Fig 3A and 3B, and S2E Fig. Paired t-tests were used for analyses in Figs 4B–4D, 4F, 5A–5C, 5E, 7D–7F and S6. Paired ANOVA with post-hoc Holm-Sidak method (to correct for multiple comparisons) were used in Figs 5F, 7A–7C, 8E and 9E. Unpaired ANOVA with post-hoc Dunnett T3 method (to correct for multiple comparisons) were used in Figs 7G, 8B–8D, 9B–9D, S7 Fig, and S9C Fig. Multiple comparisons adjusted p-values are reported.

## Supporting information

**S1 Fig. Gating strategy in Figs 1 and 2.** A representative example of the gating strategy used for phenotyping CD56$^{dim}$ NK cells in Fig 1 and 2. First, the most stable time of acquisition was selected. Single lymphocytes were then gated and characterized as negative for aqua-blue viability dye, CD14, CD19 (in the dump gate), and CD3. Any remaining monocyte populations were then excluded by selecting HLA-DR$^-$ cells. Total NK cells were selected based on their expression of CD56 (magenta); CD56$^{dim}$ NK cells were gated separately (blue) for further analyses.
(EPS)

**S2 Fig. Alternative gating, including HLA-DR$^+$ cells for Siglec-9 expressing CD56$^{dim}$ NK cells. (A)** Gating strategy focused on single lymphocytes, selected as dump$^-$ CD3$^-$, selection of CD56$^{dim}$ NK cells, and characterization of Siglec-9 expression. **(B)** Gating strategy focused on single lymphocytes, selected as dump$^-$ CD3$^-$, exclusion of HLA-DR$^+$ cells (as shown in original gating strategy in S1 Fig), selection of CD56$^{dim}$ NK cells, and characterization of Siglec-9 expression. **(C)** Characterization of Siglec-9 expression on dump$^-$ CD3$^-$ HLA-DR$^+$ CD16$^+$ cells. Plots show a representative example from an HIV- donor. **(D)** Percentage of Siglec-9$^+$ CD56$^{dim}$ NK cells following gating strategy shown in (A), and as compared to HLA-DR$^-$ CD56$^{dim}$ NK cells (as shown in Fig 1D). **(E)** Spearman correlation between the frequency of Siglec-9$^+$ CD56$^{dim}$ NK cells when excluding or including HLA-DR$^+$ cells.
(EPS)

**S3 Fig. Siglec-9$^+$ CD56$^{dim}$ NK cells express CD7.** Fresh human PBMCs from an HIV-negative donor were stained for CD3, CD56, CD7, and Siglec-9. The majority of Siglec-9$^+$ CD56$^{dim}$

NK cells express CD7, decreasing the chance that these cells are monocytes.
FMO = Fluorescence Minus One.
(EPS)

**S4 Fig. Gating strategy in Fig 3B.** A representative example of the gating strategy used for the phenotyping of CD56<sup>dim</sup> NK cells in Fig 3B. Single lymphocytes were then gated and characterized for viability and CD3. NK cells were selected based on their expression of CD56; CD56<sup>dim</sup> NK cells were gated separately for further analyses.
(EPS)

**S5 Fig. Infection of HUT78 and CEM.NKR cells with HIV. (A)** A representative example of HUT78 cells infected with HIV-1 SF2. Cells were analyzed for intracellular p24 by staining with anti-p24 RD1 antibody. **(B)** Cell surface Siglec-9 ligand expression. Equal number of indicated cells were incubated with varying amounts of recombinant human Siglec-9 Fc protein. The binding of Siglec-9 Fc to cells was measured using PE anti-human Fc fluorescent secondary antibody. **(C)** A representative example of CEM.NKR cells infection with HIV-1 DH12. Cells incubated for 72 h on RetroNectin-coated dishes with immobilized HIV were analyzed for intracellular p24 by staining with anti-p24 RD1.
(EPS)

**S6 Fig. Siglec-9<sup>+</sup> CD56<sup>dim</sup> cells exhibit lower cytotoxicity towards K562 cancer cells compared to Siglec-9<sup>-</sup> CD56<sup>dim</sup>.** Cytotoxicity was assessed using NK degranulation and IFNγ production (E: T = 4:1). Total NK cells were gated on Siglec-9<sup>+</sup> or Siglec-9<sup>-</sup> CD56<sup>dim</sup> NK cell subsets. Assays from each donor (4 donors were tested) were done in multiple replicates (3 replicates per donor), and the average of these replicates was used for analysis. Statistical analysis was performed using paired t-tests.
(EPS)

**S7 Fig. Siglec-9 blocking antibody decreases viral infectivity of the supernatants of co-cultures of NK and autologous HIV-infected primary CD4<sup>+</sup> T cells. (A)** A schematic representation of the workflow to evaluate the cytotoxic potential of NK cells against autologous HIV-infected CD4<sup>+</sup> T cells in the presence of Siglec-9 antibody. CD4<sup>+</sup> T cells were isolated from fresh PBMC and exposed to HIV-1 for 72 h. On the third day, effector NK cells were isolated from PBMC of the same donor and co-cultured with autologous HIV-infected CD4<sup>+</sup> T cells for 48 h in the presence or absence of Siglec-9 antibody. Following incubation, the infectivity of the supernatant from each well was evaluated on TZM-bl cells. **(B)** Data from donor 1. **(C)** Data from donor 2. **(D)** Data from donor 3. Assay from each donor was performed in 4 replicate wells (E:T 10:1; n = 3 donors). Statistical analyses were performed using unpaired ANOVA with post-hoc Dunnett T3 method (to correct for multiple comparisons) comparing the Siglec-9 Ab treated condition versus the isotype control-treated condition.
(EPS)

**S8 Fig. Produced bNAbs bind to HIV-infected cells.** Representative examples of staining HUT78 HIV-negative and HUT78/SF2 HIV+ cells with 3BNC117, PGT151, and NIH45-46. PE-fluorescent anti-human Fc secondary antibody was used for detection using flow cytometry.
(EPS)

**S9 Fig. Desialylation of HIV-infected target cells potentiates NK cytotoxicity. (A)** p24 analysis of HIV IIIB-infected CEM.NKR CCR5+ Luc+ cells. Cells incubated for 72 h on RetroNectin-coated dishes with immobilized HIV were analyzed for intracellular p24 by staining with anti-p24 RD1. **(B)** CEM.NKR CCR5+ Luc+ cells treated with 200 nM STSia for 1 h at 37°C

were incubated with 1 µg recombinant human Siglec-9 Fc protein. The binding of Siglec-9 Fc to cells was examined using PE anti-human Fc fluorescent secondary antibody. **(C)** HIV-infected CEM-NKR CCR5$^+$ Luc$^+$ cells were treated with indicated amounts of STSia or bNAbs. Treated cells were then co-cultured with effector NK cells (E:T 10:1). Luminescence was measured as a marker of intact (unkilled) HIV+ cells. Statistical analysis was performed using unpaired ANOVA with post-hoc Dunnett T3 method (to correct for multiple comparisons) comparing all conditions against the control condition. There are four technical replicates in each condition.
(EPS)

**S10 Fig. NIH45-46-STSia conjugate promotes higher NK cytotoxicity against HIV+ cells compared to NIH45-46 alone.** Effector NK cells were isolated from PBMC of an ART-suppressed HIV+ donor (ART05) and co-cultured with a mixture of HIV-uninfected PKH26-labeled CEM.NKR CCR5$^+$ Luc$^+$ (red cells) and HIV-infected CEM.NKR eGFP$^+$ cells (green cells). Cell mixture was treated with NIH45-46, NIH45-46STSia, or isotype control. After 24 of co-culture, the Celigo image cytometer was used to directly visualize and count the number of PKH26-labeled (red) and GFP$^+$ (green) target cells. The experiment was performed in triplicate at E:T 10:1. Plot of the raw GFP+ green (HIV-infected) cell count (left y-axis) and red PKH26-labeled (HIV-uninfected) cell counts (right y-axis). The fold reduction compares the average of each condition to the cell-only condition.
(EPS)

**S11 Fig. Siglec-9 blocking antibody characterization. (A)** Binding ELISA of anti-Siglec-9 antibody. Recombinant human Siglec-9 protein was used in ELISA to assess the antigen-specific binding of the Siglec-9 blocking antibody. His-tag protein was used as a negative control. **(B)** Western blot analysis of recombinantly-expressed anti-Siglec-9 antibody. Human recombinant Siglec-9 (hSiglec-9; 1µg/mL) and human recombinant Siglec-3 (hSiglec-3; 1µg/mL) proteins (R&D Systems) were used to determine the binding of the anti-Siglec-9 antibody to these two proteins. Odyssey Protein Molecule Weight (LI-COR) was used as a standard marker.
(EPS)

**S1 Table. Clinical data of the study participants whose cells were used for the experiments in Figs 1, 2, and 3A.**
(DOCX)

**S2 Table. Clinical data of the study participants whose cells were used for the experiments in Fig 3B.**
(DOCX)

## Acknowledgments

We thank the HIV-1 patients who participated in the study and their providers; and Dr. M. Ostrowski at the University of Toronto for providing HIV-infected Viremic samples. Cells from HIV-negative donors were obtained from Penn CFAR human immunology core. We would like to thank Rachel E. Locke, Ph.D., for providing comments.

## Author Contributions

**Conceptualization:** Mohamed Abdel-Mohsen.

**Data curation:** Opeyemi S. Adeniji, Leticia Kuri-Cervantes.

**Formal analysis:** Opeyemi S. Adeniji, Leticia Kuri-Cervantes, Qin Liu, Mohamed Abdel-Mohsen.

**Funding acquisition:** Han Xiao, Mohamed Abdel-Mohsen.

**Investigation:** Opeyemi S. Adeniji, Leticia Kuri-Cervantes, Chenfei Yu, Ziyang Xu, Michelle Ho, Glen M. Chew, Costin Tomescu, Ashley F. George, Kar Muthumani, Han Xiao.

**Methodology:** Opeyemi S. Adeniji, Leticia Kuri-Cervantes, Chenfei Yu, Ziyang Xu, Michelle Ho, Glen M. Chew, Costin Tomescu, Ashley F. George, Kar Muthumani, Han Xiao.

**Project administration:** Mohamed Abdel-Mohsen.

**Resources:** Cecilia Shikuma, Nadia R. Roan, Lishomwa C. Ndhlovu, Kar Muthumani, David B. Weiner, Michael R. Betts, Han Xiao, Mohamed Abdel-Mohsen.

**Supervision:** Mohamed Abdel-Mohsen.

**Visualization:** Opeyemi S. Adeniji, Leticia Kuri-Cervantes, Mohamed Abdel-Mohsen.

**Writing – original draft:** Opeyemi S. Adeniji, Mohamed Abdel-Mohsen.

**Writing – review & editing:** Opeyemi S. Adeniji, Leticia Kuri-Cervantes, Chenfei Yu, Ziyang Xu, Michelle Ho, Glen M. Chew, Cecilia Shikuma, Costin Tomescu, Ashley F. George, Nadia R. Roan, Lishomwa C. Ndhlovu, Qin Liu, Kar Muthumani, David B. Weiner, Michael R. Betts, Han Xiao, Mohamed Abdel-Mohsen.

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
