## [Decision Letter · Decision Letter 0]

9 Jun 2021

Dear Dr Abdel-Mohsen,

Thank you very much for submitting your manuscript "Siglec-9 defines and restrains a natural killer subpopulation highly cytotoxic to HIV-infected cells" for consideration at PLOS Pathogens. As with all papers reviewed by the journal, your manuscript was reviewed by members of the editorial board and by several independent reviewers. We regret to inform you that we will not be able to accept it for publication as it is in Plos pathogens. Though the manuscript is not suitable in its current state, this manuscript could be reconsidered but with extensive revisions that takes into account the reviewers' comments. Indeed, we and the three reviewers felt that your manuscript is original, creative with novel results for an important area of investigation. However, reviewers raise a number of important issues that should be addressed carefully. In particular, we would like you address one recommendation that is common to the three reviewers, which is the use of HIV-1 infected CD4+ primary T cells to validate the data. In addition, statistics and gating strategies should be checked as requested by reviewer 1 and reviewer 2, respectively. Other points could be addressed through discussions or experiments.

We are sorry we cannot be more positive at the present time and we hope that the enclosed reviews are useful.

We cannot make any decision about publication until we have seen the revised manuscript and your response to the reviewers' comments. Your revised manuscript is also likely to be sent to reviewers for further evaluation.

Sincerely,

Florence Margottin-Goguet

Associate Editor

PLOS Pathogens

Susan Ross

Section Editor

PLOS Pathogens

Kasturi Haldar

Editor-in-Chief

PLOS Pathogens

orcid.org/0000-0001-5065-158X

Michael Malim

Editor-in-Chief

PLOS Pathogens

orcid.org/0000-0002-7699-2064

Reviewer's Responses to Questions

**Part I - Summary**

Reviewer #1: In this study, Adeniji et al. evaluate the expression of Siglec-9, a lectin with inhibitory functions, on natural killer cells in the context of HIV-1 infection. They showed that Siglec-9+ NK cells are depleted from PBMCs of HIV-1 infected individuals (both in viraemic and ART-treated patients) as compared to healthy individuals, which has not been reported previously. Using high dimensional flow cytometry, the authors show that during HIV-1 infection, Siglec-9+ NK cells displayed higher expression of activated receptors and lower of inhibitory ones as compared to their Siglec-9neg counterpart.

Interestingly, the authors also observed an inverse correlation between Siglec-9 expression of NK cells and HIV-1 DNA levels in CD4 T cells from ART-treated HIV-1 patients, similarly of what has been reported during HBV infection (PMID: 29899741).

Evaluating the functions of Siglec-9+ cells against HIV-1 infected cell lines, they convincingly demonstrate (using various assays and different cell lines) that Siglec-9+ NK cells from healthy donors have higher cytotoxic capacities than Siglec-9neg NK cells. Those cytotoxic functions were enhanced by an anti-Siglec-9 blocking antibody.

The authors further develop and make use of Sialidase coupled to HIV-1 broadly neutralizing antibodies (bNAbs) (bNAb-STSia conjugates) as a proof-of-concept to demonstrate that those constructs are more potent than bNAbs alone in enhancing NK killing of HIV-1 infected cell lines.

Those interesting results are novel and may open new ways aiming at a ‘functional cure’ against HIV-1 infection. However, the reported decrease in Siglec-9+ NK cells in HIV-1 patients together with the absence of data on the effect of the bNAb-STSia conjugates on NK cells from HIV-1 infected patients limit the significance of the study.

Reviewer #2: The study presented by the authors deals with a potential immune checkpoint molecule on NK cells and is thus of high relevance in the HIV research field and beyond that. The authors are very well known in the field of HIV for their outstanding expertise regarding glycosylation, inflammation and anti-HIV CD8+ T cell responses. Here they propose to apply their outstanding expertise and knowledge regarding glycosylation on NK cells.

This study is original for several reasons. While Siglec-7 and other Siglecs have been studied on NK cells during HIV infection, the role of Siglec-9 has not been addressed. A negative correlation was observed between HIV DNA in CD4+ T cells and the percentage of Siglec-9+CD56dim NK cells during ART. Moreover, in the 2nd part of the presented work, the authors conjugated three bNabs to Sialidase and evaluated their impact on NK cell activity against HIV-infected cells. These conjugates selectively desialylated HIV-infected cells in vitro and enhanced NK cells’ capacity to kill them.

Reviewer #3: Adeniji et al. describe a novel method to improve the ability of NK cells to target HIV-infected cells by releasing the brakes induced by Siglec-9 expression on a subset of highly cytotoxic NK cells. This manuscript begins by demonstrating that Siglec-9 is expressed on a subset of NK cells with high expression of activating NK cell receptors capable of enhancing NK cell cytotoxicity. They show an impressive inverse correlation between the frequency of Siglec-9+ NK cells and the level of the HIV reservoir, suggesting that Siglec-9+ NK cells could play a role in limiting reservoir size. In vitro, they demonstrated that Siglec-9+ NK cells had high cytotoxicity, but that Siglec-9, as an inhibitor receptor, restrains their cytotoxic function. In a very innovative approach, they devised an approach to harness NK cell activity by selectively disrupting Siglec/sialoglycan interactions by conjugating sialidase to HIV-specific bNAbs, revealing that this could enhanced the ability of NK cells to kill HIV-infected cells. This paper breaks new ground by defining a previously unappreciated pathway to enhance the ability of NK cells to kill HIV-infected cells. This is a particularly important area of investigation as various “shock-and-kill” strategies employing NK cells for the kill gain traction. This could therefore define a high-impact area, particularly given recent promising results in targeting Siglecs in the setting of autoimmunity and cancer as proof of principle for in vivo effects of this pathway. The paper takes a creative approach to a neglected area, and the approach of harnessing bNAbs to selective di-sialate cells is creative and interesting. However, there are several concerns that should be addressed to assure the specificity and impact of the findings. The most critical concern is that the authors do not demonstrate that primary HIV-infected CD4+ T cells express sufficient sialoglycans to have the siglec/sialoglycan axis play a role in suppressing NK cell killing in the real world. If this is just an observation in cultured cells that express high levels, then it is of limited utility. Using more relevant HIV strains (e.g. T/F strains rather than lab-adapted) and infected cells (primary CD4+ T cells) is imperative to fully establish this as an important pathway. Additional demonstration that the observed effects are not due, at least in part, to monocytes, should also be addressed. Finally, the very small number of replicates (and resulting inapproapriate statistics) is concerning and undermines confidence for many of the major findings in Figures 4-6.

**Part II – Major Issues: Key Experiments Required for Acceptance**

Reviewer #1: The main limitation of the study is the use of NK cells isolated from healthy donors in their in vitro killing assays against HIV-1 infected cell lines. As pointed by the author (line 289), it would be important to validate the effect of the bNAb-STSia conjugates observed on HIV-1 infected cell lines using autologous HIV-1 infected CD4+ T cells.

The use of NK cells isolated from HIV-1 infected individuals in these setting would also be important, especially as the authors report that Siglec-9+ NK cells are depleted in those individuals, thus possibly limiting the effect of the bNAb-STSia conjugates.

Furthermore, whether sialidase impact the effector functions of the anti-gp120 bNAbs directly as previously pointed out by the authors (PMID: 30707400) should be tested. Alternatively, blocking Fc receptors (CD16 on NK cells) in the cytotoxic assay would be important to further confirm the direct role of sialic acid in dampening NK cytotoxic responses.

Reviewer #2: The gating strategies raise some questions. For example, on Figure 1A it seems that CD8+ cells have been excluded (dump CD3-CD8-DR-). However, a large proportion of the NK cells in human blood expresses CD8. Is this a typo or not a representative example of what has been done?

Also, while in the past, many studies indeed excluded HLA-DR for NK cell gating, it is nowadays well known that NK cells can express HLA-DR. Of note, stimulated NK cells, for example through IL-2, express more often HLA-DR and can display enhanced activity against pathogens. For Siglec 7, it has been described that it becomes inhibitory on NK cells once the latter get activated. Thus, excluding HLA-DR from the gated NK cell population, might exclude a particularly interesting NK cell population to study in the analyses that used the gating strategy shown in Suppl. Fig1A (ie Figures 1,2,3A-C). Indeed, HLA-DR+ NK cells have been reported to be increased in HIV controllers (doi: 10.1073/pnas.1302090110) Another gating strategy, including the DR+ NK cells, might modify some of the conclusions of the work presented here.

Figure 1B: Were the analyses performed in random order between donors from the distinct groups or was each group analysed separately? Were the antibody concentrations and/or gatings homogeneous in all experiments and between the groups?

The conclusions made for Figure 1 is a depletion of the Siglec-9+CD56dim NK cells in HIV infection. Other interpretations are also possible. Given the gating strategy, it is not excluded that they are not depleted but got activated and up-regulated DR, which would exclude them from the analysis. Only percentages are shown. Since the data are in blood, is it possible to show also the absolute numbers? A decrease in frequency can thus be a reflection of the increase of another population or also be due to a modulation of the phenotype (in chronic HBV infection, Siglec-9 expression is decreased).

Fig.3.D. It could be interesting to have more detail about the subsets that correlate with HIV DNA during ART. The text says that total CD56dim Siglec9+ NK cells correlated with HIV DNA but that Siglec7 did not (data not shown). This is intriguing since the data show that siglec9+ NK cells are those that also express high levels of Siglec7 (Fig2B). Since Siglec7 is strongly expressed on the cluster with higher expression of activation markers, including Perforin, it could mean that the NK cell subpopulation that is decreased in HIV ART, is not the one with the higher functional activity. It could thus be interesting to also show the correlation with the other markers expressed by Siglec9+ NK cells or between the clusters they identified in the Figure 3 and the HIV DNA.

Why cells from only three donors where tested in Fig.4C and Fig.4D, while in the Suppl.Fig.3, more donors were used. The degranulation and IFN-g activity shown in these graphs are close to the threshold of statistical significance. Is this sufficient to conclude about the difference with regard to target cell specificity? Does the expression of Siglec-ligands on the K562 cells could explain the difference seen between the target cells ?

Fig.5. This approach is not specifically directed against Siglec-9, which doesn’t diminish the interest of the results. For the ability of the three conjugates to kill HXB2-infected CEM.NKR CCR5+ Luc+ cells, non-conjugated bNabs were used as negative controls. What about using bNabs conjugated to something else as negative control ? What would be the reduction of infection cells and cyotoxicity in that case?

Fig.5: A confirmation of the functional assays using primary cells and primary HIV isolates would support the data.

The Siglec-9+ CD56dim NK cells are, in concordance with other studies in the literature, corresponding to a more activated, cytotoxic and mature phenotype (Figs 1-3). To which NK cell population described in other HIV/SIV studies do these Siglec-9+ CD56dim NK cells probably correspond the most? In other words, are the dynamics of these NK cells with regard to activation and maturation state, and association with viral load, showing similarities with other mature NK cell populations described in other studies on HIV/SIV infections ? This could be developed in the discussion part.

Reviewer #3: 1. A major concern about the broader applicability of the results comes from the fact that the experiments were largely done in an experimental system using HuT78 cells that have high levels of Siglec-9 ligands compared to primary T cells. Without demonstration that Siglec-9 engagement dampens NK cell responses to HIV-infected primary CD4+ T cells, it is unclear whether this observation is relevant/important to HIV pathogenesis.

2. Another concern is the selection of viruses used: SF2 (dual tropic) and other lab-adapted strains. In this era where a variety of T/F and other primary viruses are readily available, it is important to demonstrate that this is relevant in ‘real-world’ viruses.

3. Better explanation of the gating scheme for the characterization of the NK cell population is needed for Figures 1 and 2, and possible adjustments may be required. The overarching concern is this: many times monocytes can appear in NK cell gates because monocytes, particularly after activation, can upregulate CD56 (for example, Milush et al. Blood 2009 PMID 19805616 and Krasselt et al 2013 PMID 24286519). Further, as Siglec-9 is expressed on a relatively small subset of NK cells, but is somewhat universally high on monocytes, this raises a real concern as to whether some of the Siglec-9+ NK cells could actually be monocytes. This seems unlikely in light of the co-expression of NKp30 and other more NK-cell specific markers, but many times these markers can be somewhat fluid in their definition, and care needs to be made to assure that monocytes are not explaining the results. For instance, some assays that were thought to be specific to NK cells for ADCC might actually represent activity of monocytes, and this is a very important consideration (Kramski et al., 2012 PMID 22841577). Specific concerns/questions about the gating include:

a. Based on the Milush paper above, it would be ideal to include CD7 in the gating strategy as this should more definitively exclude monocytes as monocytes do not express CD7. Expression of CD7 can be more variable on CD56bright NK cells, but as this study focuses on CD56dim cells that should not present a problem. It is understandable if samples from the same cohort cannot be restained, but at least some healthy controls should be stained with a panel inclusive of CD7 to assure that inclusion of this gate does not eliminate the Siglec-9 expressing cells, because, in fact, some are monocytes. With regard to this it is important to note that not only can monocytes upregulate CD56, but that they can also downregulate HLA-DR (Abeles et al, Cytometry A 202 PMID 22837127). In particular, looking at the HLA-DR gate, it looks like there may be a low, mid, and high, and the gate may cut through the mid, thus including potential monocytes in the gating strategy.

b. Unclear why CD8 was a negative gate, as a subset of NK cells do express CD8, and in fact, increase in CD8+ NK cells is associated with lower HIV disease progression (Ahmad et al 2014 PMID 25122796). By gating out this population of highly mature and responsive NK cells, the authors could have enriched for a more rare subset of responding cells with Siglec-9, if they had, in fact, removed most of the HIV-responsive cells.

**Part III – Minor Issues: Editorial and Data Presentation Modifications**

Reviewer #1: Fig2A. The tSNE representation of concatenated Siglec-9+ CD56dim cells from HD, ART+ and viremic HIV-1 patients does not allow to identify neither clear clusters, nor differences between those 3 groups. Would a tSNE/UMAP representation of total CD56dim NK cells give better discrimination of NK clusters or differences between the groups?

Fig2C. The choice of the illustrating dot plots is questionable, not displaying representative samples but instead the ones with the highest difference between Siglec-9+ and Siglec-9neg populations, possibly misleading the readers. The authors also focused on the description of Siglec-9+ vs. Siglec-9neg in HIV-1 patients; it would be important to describe the differences of NK phenotype between controls and patients irrespective of Siglec-9 expression (e.g. expression of CD38, CD161, NKp30, Siglec-7, TIGIT).

Fig3. The FlowSom analysis should be described in the material and methods section, with mention on how the number of clusters was pre-determined. Indeed, the tSNE representation (Fig3A) does not support the division in 8 different clusters, as only cells from cluster 5, 6 and 7 cluster together, while most cells are mixed, thus questioning the use of this analysis and the conclusions drawn from it.

Fig3B. The MFI values represented in this heatmap do not correspond to the MFI reported in Fig2C. This heatmap does not show clear differences in marker expression between the different clusters, furrther questioning the cluster definition.

Figure 4. How the E:T ratio were set up for the different assays?

Material and Methods: please describe the generation of in-house Siglec-9 Ab and how it was validated.

Fig S1B. Check the axis of the first dot plot.

Fig 1D. Check y axis legend.

Fig 2A. Colour legend missing.

Line 89. Supplementary Figure 1B (instead of Supp. Figure 2).

Fig 6. Please indicate the E:T ratio. The statistics should be describe further, i.e. naming the test used for the multiple comparisons.

Reviewer #2: Line 79, line 82-83: Clusters 5 and 7 are decreased in viremic, not in ART (line 79).

Fig 3: The sum of clusters was not close 100%. This could probably easily be explained by many little subpopulations that are not part of any cluster. Add more explanations on Fig.3A.

The number of donors should be mentioned for each figure (for ex Fig 4B, the text says “several” donors, line 100).

Fig. 4D on line 115 probably means Figure 4B.

Reviewer #3: 1. It would be helpful throughout the manuscript if the n was listed in the figure legend.

2. Assuming the caveats above are taken care of (ie that monocytes in the NK cell gating scheme are not driving results), the inverse relationship between the frequency of Siglec-9+ NK cells and HIV reservoir is impressive. Frankly, it is unclear what additional information, if any, is derived from the clustering analysis in Figures 3A-C as they do not significantly add to the conclusions.

3. Overall, the statistics in this paper are sorely lacking. In many cases, a parametric test is used with very small n (t-tests used repeatedly with n=3). Parametric tests require normally distributed data, but it is impossible to ascertain normality of data with such a small n.

4. The data presented in Figures 4-6 present a compelling and novel story that Siglec-9+ NK cells are enhanced in killing HIV-infected cells and that this activity can be further enhanced by blocking the ability to Siglec-9 to interact with its ligands and dampen NK cell activation. The fact that purified NK cells were used lessens the concerns raised above about contaminating monocytes (yet not fully given the explanation above in #1). However, there are a few concerns that dampen confidence in these findings. First, related to the statistics above, most experiments were done with only 3 donors (except Figure 6D-F which used 4, likely because one of the four doesn’t show the desire effect?). As these experiments use healthy donor NK cells, this seems an unreasonably small size at it seems reasonable to ask for a few more replicates to assure that this is a reproducible effect given heterogeneity in NK cell populations between donors.

5. The quite interesting results in Figure 4D-F rely on an antibody to block Siglec-9. Validation of this antibody should be provided either through reference in the supplement. This could be done by evaluating SHP-1 phosphorylation.

PLOS authors have the option to publish the peer review history of their article (what does this mean?). If published, this will include your full peer review and any attached files.

Reviewer #1: No

Reviewer #2: No

Reviewer #3: No
---

## [Decision Letter · Decision Letter 1]

13 Oct 2021

Dear Dr Abdel-Mohsen,

We are pleased to inform you that your manuscript 'Siglec-9 Defines and Restrains a Natural Killer Subpopulation Highly Cytotoxic to HIV-infected Cells' has been provisionally accepted for publication in PLOS Pathogens.

Best regards,

Florence Margottin-Goguet

Associate Editor

PLOS Pathogens

Susan Ross

Section Editor

PLOS Pathogens

Kasturi Haldar

Editor-in-Chief

PLOS Pathogens

orcid.org/0000-0001-5065-158X

Michael Malim

Editor-in-Chief

PLOS Pathogens

orcid.org/0000-0002-7699-2064

Reviewer Comments (if any, and for reference):

Reviewer's Responses to Questions

**Part I - Summary**

Reviewer #1: The authors added novel data to their manuscript, strengthening their findings, that

1) Siglec-9pos CD56dim NK cells may be important during HIV-1 infection (inverse correlation with HIV-1 RNA viral load in viremic and with DNA viral load in ART-treated HIV-1+ individuals).

2) Siglec-9pos CD56dim NK cells have higher cytotoxic capacities against HIV-1 infected cells as compared to their Siglec-9neg counterpart (now using also HIV-1 infected primary CD4+ T cells).

2) Sialidase coupled to HIV-1 broadly neutralizing antibodies (bNAbs) (bNAb-STSia conjugates) increase the cytotoxic capacities of NK cells from both healthy individuals and HIV-1 infected persons.

Major and minor comments have been addressed thoroughly.

**Part II – Major Issues: Key Experiments Required for Acceptance**

Reviewer #1: (No Response)

**Part III – Minor Issues: Editorial and Data Presentation Modifications**

Reviewer #1: (No Response)

PLOS authors have the option to publish the peer review history of their article (what does this mean?). If published, this will include your full peer review and any attached files.

Reviewer #1: No

---

## [Editor Report · Acceptance letter]

21 Oct 2021

Dear Dr. Abdel-Mohsen,

We are delighted to inform you that your manuscript, "Siglec-9 Defines and Restrains a Natural Killer Subpopulation Highly Cytotoxic to HIV-infected Cells," has been formally accepted for publication in PLOS Pathogens.

Best regards,

Kasturi Haldar

Editor-in-Chief

PLOS Pathogens

orcid.org/0000-0001-5065-158X

Michael Malim

Editor-in-Chief

PLOS Pathogens

orcid.org/0000-0002-7699-2064